# KarstConduitCatalogue: a dataset of LiDAR derived point clouds for the analysis of karstic conduit geometry and morphology

Tanguy Racine[1], Celia Trunz[1], Julien Straubhaar[1], Stéphane Jaillet[2], and Philippe Renard[1]

[1]Centre for Hydrogeology and geothermics, University of Neuchâtel, 11 Rue Emile Argand, 2000 Neuchâtel, Switzerland
[2]EDYTEM, Université Savoie Mont-Blanc, France

**Correspondence:** Tanguy Racine (tanguy.racine@unine.ch)

**Abstract.** In mature karst aquifers, networks of interconnected conduits focus and control water flow and solute transport. In order to improve the knowledge of the multi-scale geometry of typical conduits, we acquired a data set of point clouds and triangulated surface models of over 20 different underground caves: KarstConduitCatalogue (Racine et al., 2025a, available at https://doi.org/10.60544/sbjr-z851). We employed terrestrial and mobile laser scanning workflows as fast and reliable methods for acquiring a dense point cloud of wall surfaces in enclosed spaces. These collected data can be used for many different purposes: evaluation of geometrical descriptors, direct numerical simulations of flow and transport, geomorphological mapping, structure and fracture mapping etc. In this paper, we present the various assets derived from the acquisition. The conduits presented herein span a variety of karst massifs of Western and Central Europe, from low-elevation karst plateaus to higher-elevation Alpine aquifers.

## 1 Introduction

Obtaining fast, accurate, and high resolution geometric information about real world objects is a prerequisite to answer many open scientific questions. Using an active sensing method like laser scanning, surveyors are able to characterise the geometry of objects by evaluating the position of many discrete sample points from the real surface. Using this set of points, the underlying surface may be reconstructed and its geometric properties quantitatively analysed. For instance LiDAR- (Light Detection And Ranging) derived data products may be used to build high-resolution digital elevation models (DEM), allowing detailed topographic analyses to be carried out. Repeated LiDAR acquisitions over several epochs allow for change detection and quantification, shedding insight into riverine erosional processes (Lague et al., 2013), mountain glacier accumulation or ablation dynamics (Réveillet et al., 2021), and sediment transport (Feagin et al., 2014). As a result, the use of laser scanners to map specific landforms at a range of scales and monitor their change over time, has become ubiquitous in geosciences.

Spurred with the advent of increasingly powerful processors and the miniaturisation of the sensors, the use of laser scanners to measure high resolution 3D geometries of cave passages or chambers has accelerated in the last two decades (Idrees and Pradhan, 2016). The underground environment presents however an inherent challenge in the form of limited line of sight in cave or mine passages. General cave passage tortuosity, dissolution morphologies and secondary mineral deposits all result in numerous and complex occlusions or gaps in the acquired point cloud (Figure 3). The workflow for cartography using

terrestrial laser scanner devices, which operate on fixed stations and acquire cave geometry data by a full rotation of the sensor, relies on the operator's choice of fixed stations to guaranteeing enough overlap between each scan to allow for accurate co-registration, as well as enough coverage of complex shapes (Gallay et al., 2016). Mobile mapping using handheld devices with live user feedback largely overcomes these challenges by allowing the scanning operator to multiply the viewpoints of the active sensor (Bosse et al., 2012). Finally, the problem of segmenting and classifying the datasets have been solved by calibrating the

scanner return intensity to classify contrasting lithologies (Nováková et al., 2022). The collection, manipulation, visualisation, and interpretation of dense 3D point clouds is now possible on even moderately powerful desktop computers. This has opened the doors to an increasing number of investigations dealing with surface reconstruction of artefacts or speleogenetic features in caves, as well as more detailed queries on the spatial distribution and relative chronology of sedimentary deposits and the orientation of structural features.

At its core, LiDAR-based telemetry is suited to the dark underground environment affords faster acquisition and post-processing times than visual methods like Structure-from-Motion, while the latter provides a strong alternative in terms of accuracy, feasibility and cost-effectiveness (Giordan et al., 2021). Since despite its cost however, LiDAR telemetry overcomes many challenges inherent to light-based techniques, the use of terrestrial laser scanners (TSL) in low-light underground environments has become a standard for detailed geometric reconstructions (Idrees and Pradhan, 2016), with mobile mapping

solutions also being increasingly explored (Dewez et al., 2016a, e.g.,). Lidar scans are digital twins of cave site, in the form of high resolution point clouds or meshes. They have been leveraged by a wide range of studies bearing on documentation of archeological heritage sites (Grussenmeyer et al., 2012), speleogenetic interpretations (Gallay et al., 2016; Fabbri et al., 2017; Konsolaki et al., 2020), structural analyses and stability assessments (Idrees and Pradhan, 2018; Kazmierczak et al., 2020), improving show-cave management (Milius and Petters, 2012; Pfeiffer et al., 2023), or detailed and accurate cartography

(Šupinskỳ et al., 2022). Long term campaigns to document complex cave systems developed over more than 10 km are well underway all over the world (Kaňuk et al., 2024).

Lasergrammetric surveys have been leveraged underground to support structural and speleogenetic interpretations in varied contexts. Hajri et al. (2009) used a TLS derived point cloud to reconstruct a dense mesh and, through automated classification, investigated the relation between geometric parameters of a stalagmite forest and their proximity to an underlying karst

conduit undermining their structural integrity. De Waele et al. (2018) reconstructed the spatial and relative temporal relationships between ceiling dissolution morphologies using a combination of TLS and photogrammetric approaches. Elsewhere in a Pyrenean cave, under favourable geological settings offering strong lithological contrasts, Nováková et al. (2022) developed a complex workflow leveraging a LiDAR point cloud's intensity and colour attributes to classify different bedrock types within the underground environment. One of the more complete uses of a high resolution laser scan in an Italian cave by Fabbri

et al. (2017) showcases how cm to dm size morphologies can be observed and added to a geodatabase of speleogens, allowing the various phases of speleogenetic development to be distinguished both spatially, in a quantitative sense, and temporally, in a relative sense. Finally, repeated high resolution acquisitions open the door for yet other applications dealing with change detection. Sediment or ice mobility in the underground environment was evidenced through multi-temporal LiDAR and photogrammetric projects tracking moving targets or differencing series of surfaces (Blatnik et al., 2023; Šupinskỳ et al., 2019;

Securo et al., 2022). Therefore, survey approaches resulting in detailed and spatially accurate geo-databases of underground objects are extremely valuable for morphometric analyses, speleogenetic reconstruction and change detection. However, most of these datasets are not widely available to the scientific community

There is a need for new high-resolution geometric information on karst conduits to better inform the statistical metrics of cave networks because models for flow and transport rely on passage size distributions, and network topology. Existing studies
on large cave datasets, e.g., (Collon et al., 2017; Jouves et al., 2017) rely on traditional speleological data, containing relatively sparse information of passage size at each measured station. Here we demonstrate that the speleological measurements can be supplemented by the detailed point cloud data. We propose a workflow for extracting a curve skeleton (Cao et al., 2010) or 3D-centreline from the point cloud, and investigating the geometric properties of the karst conduit along this 3D curvilinear object.

The aim of the work presented in this paper was to acquire and share cave scans covering a broad range of hydrologically active conduit morphologies ranging from phreatic to vadose (Lauritzen and Lundberg, 2000). The aim of this dataset is not to provide complete scans of existing cave systems but to sample a broad spectrum different conduits considered as hydrologically consistent units. We chose the scan sites foremost due to their hydrological function: ranging from periodically flooded to inactive / relict stream passages. We also gave consideration to the ease of access and scanning by mobile mapping, rejecting
too-narrow or too-vertical cave sections which were impractical to scan. We then explored as many lithology and structural setting types as possible within the project time-frame and the broader alpine geographic area. The cave scans come from various karst massifs of the Jura, the European Alps, the French central Massif, and the Classical Karst of Slovenia (Figure 1). This allowed the conduits catalogued hereafter to span a range of hydro-geologic, lithological, and structural settings, with varying degrees of sediment fill and secondary mineral deposition.

The data set is available through the *KarstConduitCatalogue* repository (Racine et al., 2025a). A brief overview of each passage morphology is given in each cave's metadata file. Hydrologists may find this dataset suited for the analysis of key geometric characteristics shared by typical cave conduits, including downstream distribution of apertures and roughness elements. Moreover, karst geomorphology studies benefit from high fidelity and high resolution geometric data for the georeferencing of key erosional markers in caves. This dataset also has a didactic vocation as it presents geometries of an exemplary, character-
istic nature. The spatial arrangement of various speleogenetic forms and secondary infill or deposits may be discussed as part of teaching material. The dataset can be readily viewed at lower resolution on the web-based application Potree, through the *KarstConduitCatalogue-Potree* repository (Racine et al., 2025b).

In the following sections, we present the dataset acquisition and consolidation procedure, going from the raw field data to the ready-to-use end products. For each cave passage, these products comprise 1) a georeferenced, classified point cloud, 2) a
triangulated mesh surface, 3) a simplified centreline representation and 4) 2D raster images of both floor and ceiling cutouts. We detail the pre- and post-processing steps involved in the cleaning, classification and georeferencing of the dataset and describe the various data records available. We finally showcase the end-products using a single karst conduit, Markov Spodmol cave (Slovenia) as an example.

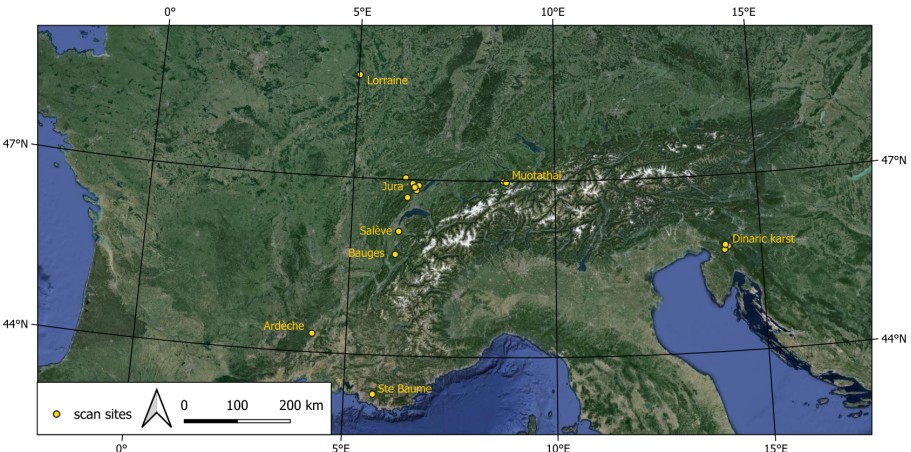

**Figure 1.** Situation map of the selected cave sites centred over the European Alps and the respective karst massifs in which the conduit scans were collected. Maps data: Google, Landsat / Copernicus, Data SIO, NOAA, U.S. Navy, NGA, GEBCOGeoBasis-DE/BKG (© 2009), Inst. Geogr. Nacional

## 2  Methods

The cave conduits were scanned with two different instruments: we used the Leica BLK2GO for the majority conduits we scanned ourselves, while at two locations, we scanned the conduits using the FARO Focus 3D instrument. We performed most of the visualisation and processing of point cloud and mesh elements using the open source and free software CloudCompare (Girardeau-Montaut et al., 2016), as well as a wrapper written in Python language (CloudCompare, 2024) to automate some of the processing routines.

Here we describe the properties for those two instruments that were used in this study, as well as the methods we used for scanning the cave and post-processing the point cloud dataset (Figure 2). We also briefly discuss the expected sampling density, and resulting resolution of the cave features which could be obtained.

### 2.1  Laser scanning of caves

#### 2.1.1  Terrestrial laser scanning

The cave passages in Rupt-du-Puits and Grotte de la Madeleine were scanned with the terrestrial laser scanner FARO focus 3D. The sensor has a range from 0.6 m to over 100 m and a ranging error of 2 mm. The point positions are recorded in polar coordinates during the distance measurement and are subsequently converted to a local cartesian system. Since the scanner sensor revolves once around a vertical axis from a fixed position, the operator usually starts the scan remotely from a hidden location and repeats the procedure to eliminate occlusions from the acquired scene.

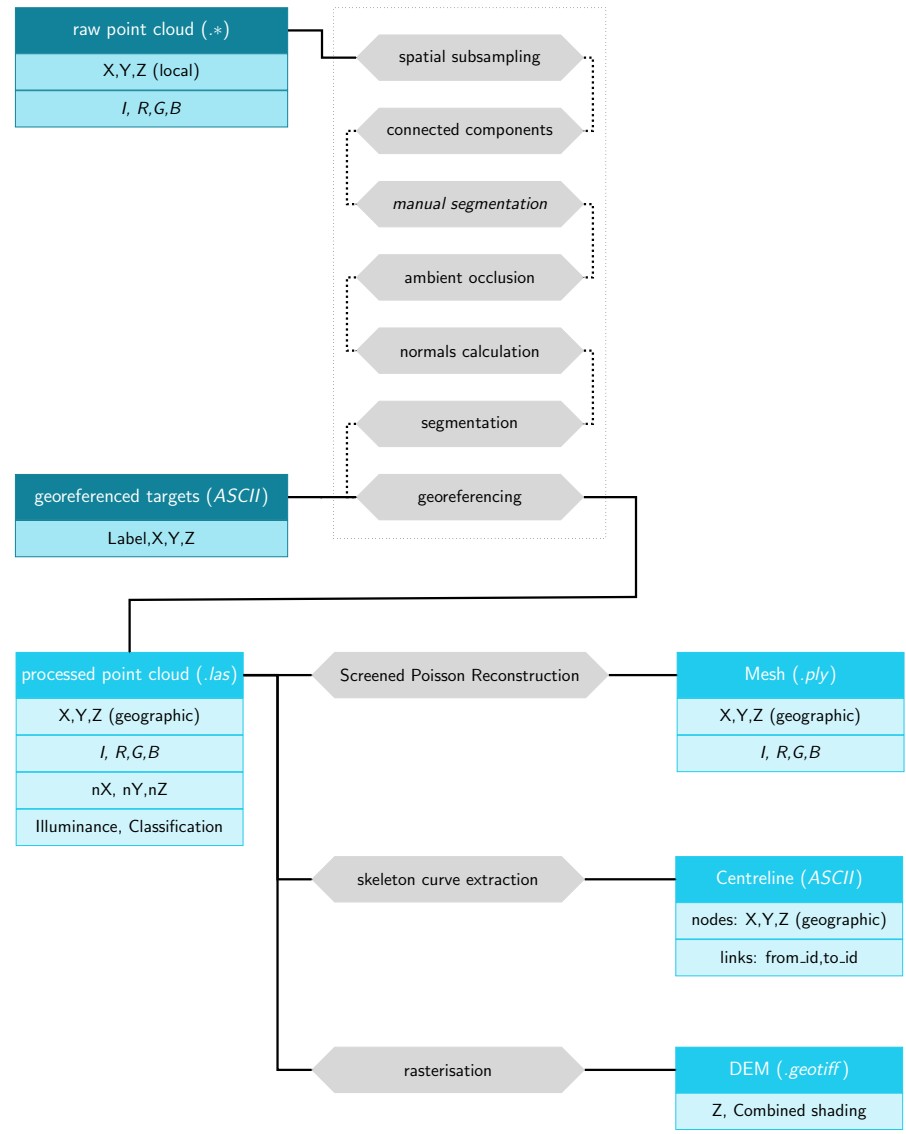

**Figure 2.** Summary of the point cloud processing workflow (grey) and delivered data records (light blue)

### 2.1.2 Mobile mapping with Leica BLK2GO

We acquired most of the cave conduit from the dataset with a light-weight mobile handheld laser scanning system (Leica BLK2GO) capable of capturing detailed point sets within an underground cavity. With the aid of 830 nm wavelength laser pulses, the scanner measures up to $420\,000\ \mathrm{pts \cdot s^{-1}}$ with a field of view (FOV) spanning $360°$ horizontally and $270°$ vertically. The sensor range goes from about 0.5 m to 25 m. The device is also equipped with a 3-camera system, each with a 4.8 Mpx sensor and $300° \times 135°$ FOV. The range error reported by the manufacturer for indoors use is $\pm 3$ mm.

At the core, mobile mapping devices consist of a LiDAR distance sensor, coupled with inertial sensors (Bosse et al., 2012; Zlot and Bosse, 2014). Assuming that the scanner's surroundings neither move nor deform, the Simultaneous Localisation and Mapping (SLAM, Bailey and Durrant-Whyte, 2006) algorithm allows for the x, y and z coordinate tuples to be stored in a local cartesian reference frame. To achieve this, the algorithm uses regular updates to the scanner position by 1) using the device's Internal Motion Unit (IMU) and 2) by triangulating between recognisable point features.

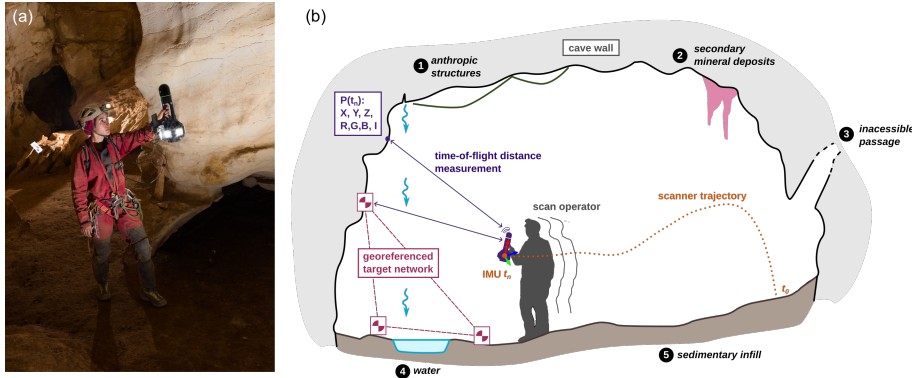

**Figure 3.** (a) Typical scanning stance of a speleologist in a cave passage, with a laminated scan target in the background (photo: Tanguy Racine) (b) Mobile cave scanning workflow and usual causes for masks, occlusions or missing data on bedrock cave walls (adapted from Racine et al., in press).

### 2.1.3 In-cave scanning strategy

The in-cave scanning workflow begins with a reconnaissance of the conduit to be scanned, identifying the various obstacles to progression. These obstacles include: passage intersections, large changes in average section dimensions, floor-steps, pits, narrow sections, waterways, etc. An overview of the acquisition progress in the form of a rough point cloud visualisation is transmitted to the scan operator navigating the cave conduit or chamber in real time over a wireless connection, facilitating decision making for an optimal scan trajectory and cave features coverage. We split the conduit to be scanned in several overlapping acquisitions (scenes) acquired separately. The decision to stop an ongoing acquisition was almost always chosen with respect to 1) encounter with an obstacle or 2) scan duration exceeding a chosen threshold. The obstacle criterion is self explanatory. The scan duration criterion was chosen with respect to the hardware specification of the phone on which the point

cloud scanning progress was displayed to the scan operator. In practice, the monitor displaying the scan progress would become unresponsive after 5-7 min, so shorter scans were preferred to better monitor any obvious gaps in the acquisition process. Using the terrestrial laser scanner, a scene corresponds to a single revolution of the scanner sensor around a vertical axis. The scan times depend on the spatial sampling density selected by the operator. For the mobile mapper, a scene corresponds to a several minutes long walk by the operator within the cave environment (Figure 3) with the scanner sensor revolving at constant angular velocity around a mobile axis. Using the mobile mapper, we scanned the conduit sections with partial spatial overlaps for subsequent co-registration. We achieved this in the field by retracing our steps anywhere between 2-10 m to guarantee that acquisitions intended to be co-registered would have enough common points.

A lighting system provided by Méandre Techologie comprising 5 LEDs with a flux of 2 250 to 15 000 lumen each, arranged around the scanner, provides near-panoramic illumination allowing for visible light information to be encoded into the point set data file as Red, Green and Blue channels. Outside the cave on a work station, we process the raw files corresponding to each acquisition scene with the proprietary software Cyclone Register 360. We carry out the co-registration of scenes in two-steps: first by visual alignment, second by iterative closest point algorithm (Besl and McKay, 1992). Finally, we export a raw, assembled point cloud to LAS format, the open and industry standard format for LiDAR data.

### 2.1.4 Point cloud density

Because of the controlled sensor rotation, TLS-derived raw clouds exhibit clear patterns of concentric rings on surfaces which were sampled only once (Figure 4a). For the mobile scanner, the instrument's constant movement results in some cave walls being more densely sampled than others (Figure 4b), and thus requires point density resampling. It is impossible to anticipate precisely the density of the final assembled point cloud. Wherever separate acquisitions are overlapping, meaning they have been visited at least twice, the point cloud has a high sampling density. For other regions, especially high in ceiling pockets or fractures, the walls are seen by the scanner only briefly, yielding a low spatial sampling density.

To harmonise the density of point coverage, we sub-sampled each point cloud using the CloudCompare spatial sampling algorithm. We set a threshold value of $d = 2$ mm and $d = 5$ cm (d being the minimum distance between a point and its nearest neighbour), for high and low resolution point clouds, respectively.

### 2.2 Georeferencing

Raw point clouds are collected in local coordinate systems. For a levelled terrestrial laser scanner, or the mobile mapper, the distance between points and position relative to the vertical are tracked by the scanner. This means that while the scale of the point cloud model is known, its overall position, and orientation relative cardinal directions need to be calculated with independently surveyed control points. To determine the necessary rigid (rotation and translation) transformation, we measured the geographic coordinates of a series of reference points by placing a series of targets along the cave: these are laminated sheets containing a visual aid to determine their centre. The position of these targets in local scan coordinates was determined by using the closest point to the centre recorded on the scan, using intensities of return and RGB information as visual aids. The targets were always placed in an easily accessible location, so that the scanner itself could be brought to bear on the target within a 1

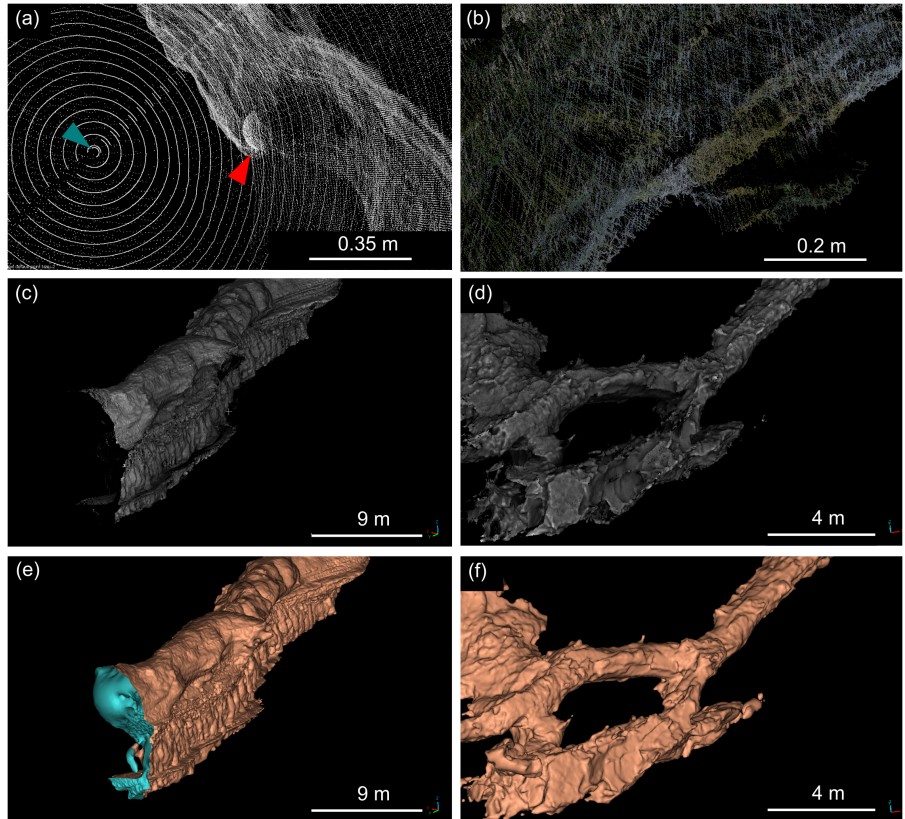

**Figure 4.** Detailed views of the scanning and meshing results using TLS (left) and SLAM (right) technologies. (a) close-up view of the raw *Rupt du Puits* point cloud with concentric circular data distribution pattern of the TLS (blue) and one of the spheres placed in the scene to help with scene co-registration (red) (b) close-up of the raw *Grotte de la Sourde* data, with overlapping, criss-crossing poin trails acquired by the BLK2GO scanner; (c) point cloud of the *Rupt du Puits*, downsampled to 2 mm and (e) the reconstructed mesh at 5 cm resolution; (d) point cloud of *Grotte de la Sourde* and (f) its reconstructed mesh.

m radius. This ensured an adequate point sampling density around the target in order to locate the target centre within less than 1 cm. We measured the position of those targets with a calibrated laser-distance meter called the disto X2 and widely used in cave surveying (Heeb, 2016). With the disto X2, we recorded the three following quantities for each shot linking two survey stations: distance, bearing and inclination. In order to control operator errors, we triplicated each shot front and back, and averaged them for each station to station shot. From these data records, we extracted the triplet of geographic coordinates for every known point in a cartesian reference frame. We compiled the resulting survey data using the public cave survey software Therion (Mudrák and Budaj, 2025), which uses the Survex (Betts, 2024) program for loop closure error calculations and shot data averaging, as well as a model to correct for magnetic declination at a given place and time on the Earth's surface.

We georeference the point clouds by calculating a rotation and translation matrix using the pairwise registration algorithm (Arun et al., 1987) between the targets' local coordinates and their geographic counterparts. This was implemented in Python

and applied to the point clouds. The new coordinates of the point clouds are calculated by matrix multiplication, applying a rigid transformation so that the Euclidean distance between pairs of points is preserved. This also serves to 1) check the validity of loop-closure whilst the SLAM algorithm is running, avoiding potential drift, and 2) to detect user blunders when assembling the scanned scenes after their acquisition.

## 2.3 Point cloud segmentation and cleaning

Dataset noise for cave scans arises from two main sources.

The first are erroneous pulse returns due to excess moisture, water droplets or interference with airborne particles. This usually results in sparse clusters of points being recorded within the cave passage itself. We use the CloudCompare algorithm to label connected components and thereby divide the point cloud in groups. With the relevant algorithm parameters we adjusted the minimum size of cluster to be labelled as a group, and the smallest pairwise distance between any two points belonging to different groups. This is an effective strategy for removing the floating clusters when choosing an appropriate minimum distance between clusters with the octree subdivision level parameter, as well as a threshold number of points defining a cluster. We were able to effectively remove noisy floating regions and solitary noise points by selecting only the clusters containing the largest number of points, which correspond to the conduit walls.

The second kind of noise in the dataset stems from occlusions or masks in the cave point cloud caused by the presence of the scan operator and or assistants. The BLK2GO scanner automatically masks out points taken in a quadrant facing the scan operator to prevent this type of self-scanning. Additionally, we minimised this type of noise with adequate scanning strategy. However narrow twisting passages often require the operator to carry the scanner in a sub-optimal orientation, putting the scan operator or any helper in the way of the laser swath. Whenever this resulted in noisy data patches (Figure 5), we removed the latter semi-automatically or manually from the cave point cloud. We adopted the multi-scale dimensionality criterion approach using the CANUPO algorithm plugin for CloudCompare (Lague et al., 2013). At the Grotte de la Cascade, we implemented this step in the workflow by labelling clusters of noisy data. In this case, the CANUPO algorithm was effective because of a critical difference in multi-scale dimensionality of noise clusters. These noisy clusters exhibit a high linearity score from the cm to the m scale) while the cave walls score highly on planarity at those scales. By segmenting out the points thus labelled as noise from the point cloud, a subsequent analysis of connected components was sufficient to remove the remaining floating clusters, negating the necessity to manually clean the point cloud . Elsewhere,manual segmentation was made on CloudCompare by iteratively selecting noisy regions and removing them from the dataset.

## 2.4 Determination of instrumental noise

The point cloud generated by the BLK2GO device has a specific 3D structure made of criss-crossing point trails which originates from the scanner movement during a survey. Following Dewez et al. (2016b), we compute the roughness distribution on a test surface (a $1.1 \times 0.8$ m whiteboard) to evaluate the performance of the scanner. First, for each point, the euclidean distance to its nearest neighbour was computed. We find that for a test surface sampled at approximately 1-2 m, the mean distance to the nearest neighbour is of 1.3 mm. We then fitted a plane to the point cloud acquired by sampling this artificial, smooth planar

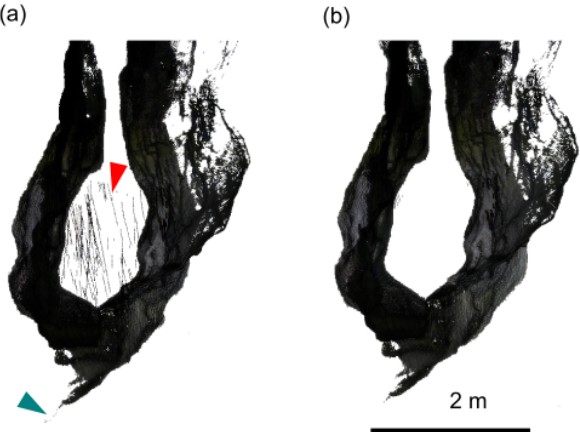

**Figure 5.** Illustration of the cleaning process (a) before automatic removal and manual segmentation of noise points, (b) after cleaning. In red: trails arising from the scan operator scanning themselves in the narrow passage. In blue: lone points corresponding to narrow fissures incompletely scanned.

surface with the BLK2GO scanner and computed the distance to this plane at each point. We find that 95% of all points fall within a distance of 0.016 m to the best-fit plane (Figure 6).

A related way in which we can confront the precision limit quoted by the manufacturer is to compute the distribution of point cloud roughness with variable neighbourhood radii, adopting the strategy of Dewez et al. (2016b). In CloudCompare, the roughness value $\sigma(r)$ can be computed at any point of a cloud and it represents the distance from a point to a plane fitted to its neighbours within a chosen search radius $r$ (Girardeau-Montaut et al., 2016). We want to find out at which scale the calculation of the implicit surface will be robust, specifically how small a neighbourhood radius one may choose to fit a surface model before instrument noise makes the reconstruction unreliable. For a simple planar surface with normally distributed noise in the normal vector direction, roughness distributions change with the search radius in a predictable way: above a given $r$, determined by instrument noise, the shape of roughness distribution should stabilise, and its parameters, like the 68th percentile, should remain constant with increasing $r$. For the BLK2GO, we find that for $r > 0.08$ m, the 68th percentile of roughness stabilises at a value of 0.01 m. This is very close to the value for 68th percentile of unsigned distances over the entire dataset. According to this result, we adopt this value of 0.08 m as the smallest possible resolution for robust normal computation and meshing steps described below (Figure 6).

## 2.5 Normals calculation and meshing

Point cloud normals are spatial vectors calculated at each point. The plane uniquely defined by the point and its normal vector is a local linear approximation of implicit surface to be reconstructed (Hoppe et al., 1992). We ran the calculation of point normals on CloudCompare by least-squares fitting of a plane surface model using the neighbourhood of each point. Here the

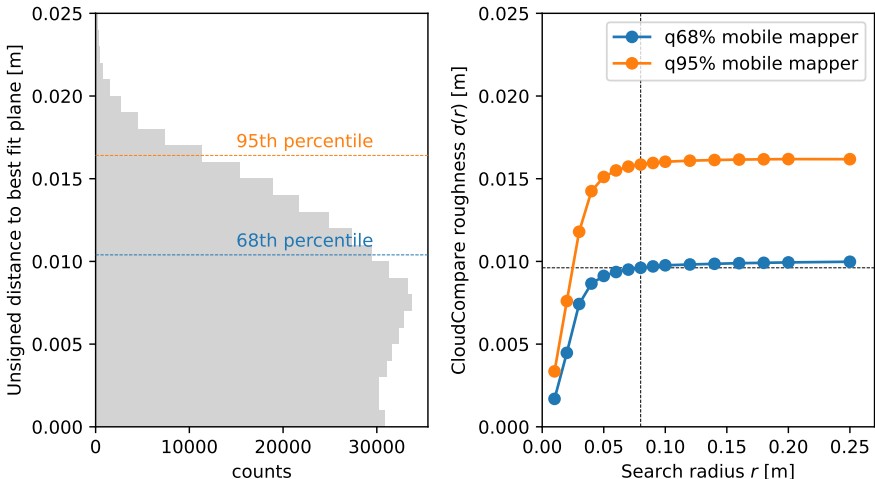

**Figure 6.** (a) distribution of unsigned distances to a best-fit plane adjusted to the test surface. The 68th percentile of this distribution ($n = 4.84 \times 10^4$ points) is taken as the position uncertainty associated with the mobile laser scanner (on the order of $0.01$ m) (b) Point-cloud roughness computed with CloudCompare as a function of the search radius: the greater the search radius, the greater the number of points used for plane fitting

neighbourhood of a point denotes the subset of points from the cloud within a specified euclidean distance, the search radius,
of that point. Following the notation of Hoppe et al. (1992), the tangent plane at the ith point is determined by its centre $o_i$ (the centroid of the point neighbourhood) and a normal vector $\hat{n}_i$. In CloudCompare, the solution normal vectors were reoriented by way of a minimum spanning tree of the k-nearest neighbours at each point; a complete description of the algorithm used to reorient the solution vectors is outside the scope of this paper but details may be found in Hoppe et al. (1992). In this graph optimisation problem, the weight of an edge between points $i$ and $j$ is taken as the scalar product between neighbouring solution
vectors, reflecting the intuition that the tangent plane at nearby points should be sub-parallel for sufficiently smooth surfaces. We started the algorithm with 6 k-neighbours and increased the number of nearest neighbours $k$ considered in the calculation of the spanning tree if at first the normals were not consistently re-oriented (Table 2). In our datasets, a neighbourhood with a search radius of $0.08$ m was found to give reliable results for finding consistently oriented normals, corresponding to the threshold at which the cave wall roughness signal drowns roughness due to instrumental limits (see Figure 6). Choosing a
smaller search radius would yield vector normal orientations affected by instrument noise. The reorientation of normals can fail at sharp boundaries of the 3D surface sampled, and it is sometimes necessary to manually segment the cloud at those edges. This is to prevent the reorientation algorithm from finding and using nearby points belonging to a different surface in the nearest neighbour search. We computed a triangulated mesh in CloudCompare, using its mesh construction routine based on point cloud normals, using a radius of 5 cm. This routine is a wrapper for the screened Poisson Reconstruction algorithm
(Kazhdan et al., 2006). Meshes were subsequently segmented and cleaned using the CloudCompare and Blender segmentation

tools to remove spuriously interpolated surfaces, for instance wherever open passage ends or areas of missing data were patched up with the algorithm. The resulting meshes are therefore not closed surfaces (Figure 4e and f).

## 2.6 Floor and ceiling raster extraction

We provide a georeferenced DEM of the cave passage floor and ceiling by classifying the cave floor and cave ceiling by
using the Cloth Simulation Filter algorithm (Zhang et al., 2016, CSF). This segmentation step is essential in most airborne LiDAR mapping campaigns, as it effectively separates ground points from non-ground points. Intuitively, a non-rigid cloth is draped over the upturned point cloud, and points touching the cloth are labelled as ground category. On a cave point cloud, this algorithm extracts ground points corresponding to the passage floor. The remaining points correspond to the cave ceiling. We considered several cases where this algorithm could classify points incorrectly and manually attributed the correct classification
to the points. Wherever gaps in the point cloud are apparent due to the presence of a water body, then only one surface (the ceiling) will appear on the scan. Running the CSF algorithm would classify the ceiling as a floor. Wherever one passage overlies another, then more than one surface should be classified as a floor. However, the lowest lying passage will hide all the others, and floor points will be mis-labelled as ceiling points. In this case, we split the point cloud into disjunct sections and subsequently ran the algorithm on each. Finally, complex floor geometries such as overhanging boulder sides will hide some
floor points from the CSF algorithm. For these cases, we manually attributed the floor attribute to the relevant points based on visual inspection.

Using the Relief Visualization toolbox (Kokalj et al., 2016), we also provide a combined image specifically designed to highlight subtle topographic changes (Kokalj and Somrak, 2019). We find that projecting the resulting point cloud as a floor or ceiling DEM and using a suitable relief visualisation techniques highlighted subtle topographic relief and can help emulate
traditional cave maps. We modified the presets for the steep Visual Archeology terrain blend, to account for the event steeper topographic features of cave floors and ceilings. The parameters used to generate the blended images (top to bottom) are given in Table 3.

## 2.7 Centreline extraction

We refer to the cave's centreline as an undirected metric graph which captures the cave conduit topology. It is based on
a subset of points belonging to the curve skeleton of the cave wall point cloud. There are many algorithms for extracting such a skeleton curve from a three-dimensional object (Tagliasacchi et al., 2016). To compute this object, we use the Python implementation of (Cao et al., 2010) point cloud contraction algorithm based on local-Delaunay triangulation and topological thinning. This technique is robust to noise and missing data, which is often present in in-cave surface acquisitions due to the common occurrence of small and / or narrow inaccessible side-passages, and water surfaces (streams, dammed pools, etc.).
There are several key parameters for this algorithm including the initial balance between the contraction and attraction weights matrices, as well as the level of downsampling of the initial point cloud on which to perform the contraction. By default, the initial contraction weights are set to 1 and the attraction weights are set to 0.5. For several cave point clouds we tested varying starting ratios of these weights and noticed that a high ratio of contraction to attraction yields fewer, rectilinear

branches than a low one. We found that the algorithm terminated within 4-5 iterations using downsampling parameters. In the end, we chose to use 0.5 for both initial contraction and attraction ratios (Table 2). The end position of the nodes describing the centreline generated by contraction follows the ratio of contraction and attraction weights defined by the user. Indeed, low attraction to contraction ratios yielded topologically simpler (fewer branches) and geometrically smoother curve skeletons, with clear differences even after the first contraction iteration (Figure C1). When using the strongest initial contraction to attraction weights ratio (Figure C1a), one can observe a strong collapse and smoothing of the point cloud geometry towards the objects skeleton. The reverse is true for the smaller initial contraction to attraction weights ratio (Figure C1d). With strong contraction, we observe that centreline nodes could even fall outside the walls of the original point cloud due to the Laplacian smoothing of the walls if the conduit bent sharply without bifurcations (see for example Figure C2).

We spatially sub-sampled the contracted point cloud to yield a sparse cloud. This sparse cloud can be thought of as a discrete sampling of the curve skeleton, i.e. the thinned 1D representation of the 3D cave wall model. We performed a final connected component analysis to remove badly contracted points from this curve skeleton cloud, as these may be located far away from the original point cloud and selected its largest component. Finally, we reconstruct the skeleton topology by considering the set of points as an undirected, complete graph, where each edge weight represents the euclidean distance between any pair of nodes. We computed the minimum spanning tree on this graph. In this mathematical object a degree-1 node is called a leaf and corresponds to a cave opening or a dead-end. Formally, the trajectory of a person or object travelling the inside of a cave conduit while avoiding its walls can be described by a walk from a start to an end node along the centreline graph. This allows us to describe variations in cave geometric properties along a given walk which corresponds to the position of an observer along the cave passage.

## 3 Data records

The general organisation of the datasets is as follows. Within the data repository, we provide one subfolder per cave, and within, one folder per elementary cave passage, following the local toponymy. Each passage folder holds a set of point clouds, meshes, rasters and centrelines, as well as the metadata file in self-describing yaml format. The repository organisation for a single cave is detailed in Figure 7. In a companion GitHub repository called pc-processing (https://github.com/ERC-Karst/pc-processing/releases/tag/v1.0.0), we also provide a set of Python scripts which we ran to 1) extract centrelines, 2) extract rasters of the floor and ceiling and 3) convert the centrelines to various formats.

### 3.1 Point clouds

Each unified, cleaned and georeferenced point cloud is archived in LAS format, the industry standard, open, binary format for interchanging point cloud data. We chose the LAS 1.4 with point format 7, which includes the RGB color channels by default. For each cave and conduit therein, we provide a point cloud spatially sampled at 2 mm and 5 cm, corresponding to high and low resolution respectively. Some datasets are georeferenced, and the point coordinates are given in their country's official coordinate reference systems. Since some of the coordinates may be very large, LAS usually provides the data using

```
root/
|--- Cave1/
|    |--- Passage1/
|    |    |--- pointclouds/
|    |    |    |--- Cave1_Passage1_sampled_2mm_PCV_normals_classified_georef.las
|    |    |    |--- Cave1_Passage1_sampled_5cm_PCV_normals_classified_georef.las
|    |    |--- mesh/
|    |    |    |--- Cave1_Passage1_mesh_5cm.ply
|    |    |--- raster/
|    |    |    |--- Cave1_Passage1_floor_4cm.tif
|    |    |    |--- Cave1_Passage1_floor_4cm_Cave_Terrain.tif
|    |    |    |--- Cave1_Passage1_ceiling_4cm.tif
|    |    |    |--- Cave1_Passage1_ceiling_4cm_Cave_Terrain.tif
|    |    |--- centreline/
|    |    |    |--- Cave1_Passage1_nodes.txt
|    |    |    |--- Cave1_Passage1_links.txt
|    |    |    |--- Cave1_Passage1_branches.txt
|    |    |    |--- Cave1_Passage1.dxf
|    |    |    |--- Cave1_Passage1.geojsons
|    |    |--- scan.yaml
|    |--- Passage2/
|    |    |...
|    |--- cave.yaml
|--- Cave2/
|    | ...
```

**Figure 7.** dataset repository structure

a Global Shift information which is stored in the data header and can be read by OpenSource programs, e.g., CloudCompare. We populate the LAS classification field with the intensity, a measure of the strength of the returning laser pulse. The data is organised in a table with headers containing a set of spatial coordinates, and additional scalar fields such as: return intensity, a triplet for red (R), green (G) and blue (B) channels and normal unit vector coordinates Nx, Ny and Nz, an integer classification flag (1: unclassified or ceiling, 2: ground), and a 64-bit float corresponding to the illuminance value which is analogous to the sky view factor (Duguet and Girardeau-Montaut, 2004), see Table 1.

**Table 1.** Point cloud data file description

| label | description | unit |
|---|---|---|
| Intensity | relative strength of pulse return, *64-bit float* | |
| Classification | point label, *integer* | |
| X, Y, Z | coordinate in cartesian geographic reference system, *64-bit float* | m |
| nX, nY, nZ | unit normal coordinate, *64-bit float* | m |
| R, G, B | red color channel intensity, *64-bit float* | |
| Illuminance (PCV) | sky view factor sampled from a sphere, *64-bit float* | |

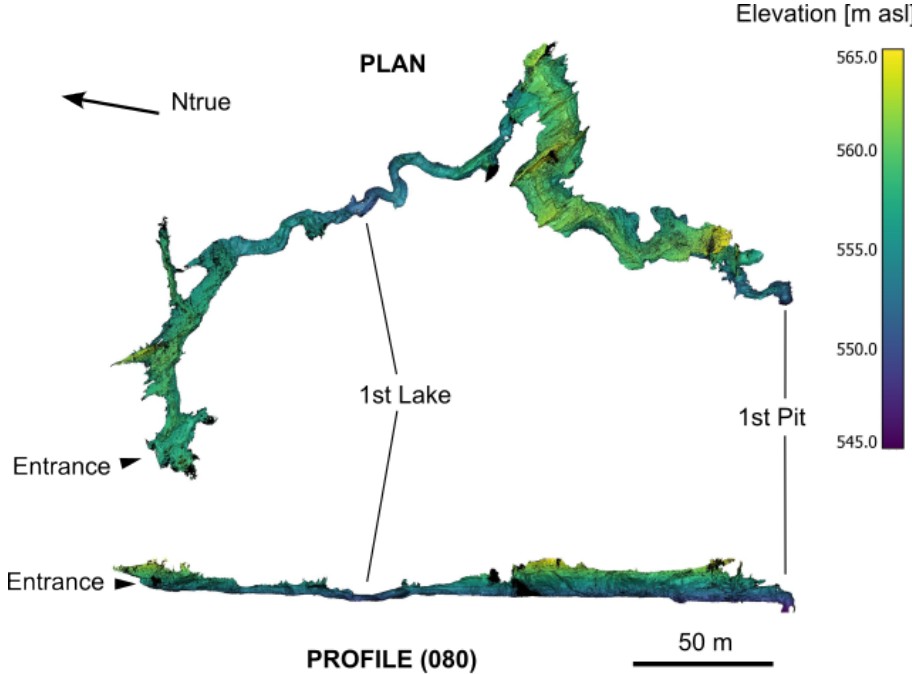

**Figure 8.** (a) plan view and projected profile of altitude coloured point cloud and (b) illuminance (PCV) coloured point cloud of Markov Spodmol cave. Ntrue denotes the orientation of geographic north.

## 3.2 Meshes

The provided meshes are the 3-dimensional representation of the cave walls. They are calculated using the Screened Poisson Surface Reconstruction algorithm (Kazhdan et al., 2006) using the parameters indicated in Table 2. The meshes are stored in binary PLY format, whereby the surface is defined by 1) a list of vertex coordinates, normals and texture information and 2) a list of faces.

## 3.3 Cave centrelines

Cave centrelines, as realisations of discrete sampling of the curve skeletons, are given as undirected graphs. These objects are stored in three ASCII files which are essentially relational tables: the first file containing the X, Y, Z geographic coordinates triplet of each point (vertex), one point per line, the line number being the unique identifier (ID) of the corresponding point. The second file contains the links (edges) between the points: each line corresponds to link between the source and target node IDs and the line number is the edge's unique ID. The third file corresponds to the branch ID for each edge: this contains two columns, first being the branch ID, the second, the edge ID. Thus one can for instance query the position of all nodes belonging to a branch by way of edge indices. All processing parameters are given in Table 2. To integrate the conduit centrelines with

the point clouds and meshes in a single visualisation, we also provide them in the interoperable Drawing Interchange Format (DXF). Further, to integrate the visualisation of the centrelines with the raster files on any Geographic Information System (GIS) software, we also provide the centrelines in the interoperable Geographic JSON (GeoJSON) format. For any given conduit, the sum of every centreline segment is given in table A1 as the surveyed length of each scan.

**Table 2.** Point cloud processing parameters

| item | software | parameter | value |
| --- | --- | --- | --- |
| spatial resampling | CloudCompare | minimum distance | 0.002 m & 0.05 m |
| connected components | CloudCompare | octree level | 10 |
| | | min. points per component | 10 |
| manual segmentation | CloudCompare | — | — |
| ambient occlusion calculation | PCV plugin | samples rays on a sphere | True |
| | | Count | 256 |
| | | Render context resolution | 1024 |
| normals calculation | CloudCompare | radius | 0.08 m |
| | | model | planar |
| normals reorientation | CloudCompare | method | Minimum Spanning Tree |
| | | k nearest neighbours | at minimum 6 |
| Screened Poisson reconstruction plugin | Screened Poisson plugin | boundary condition | Neumann |
| | | spatial reconstruction | 0.05 |
| floor extraction | Cloth Simulation Filter plugin | cloth size | 0.05 m |
| | | terrain type | steep |
| | | threshold | 0.5 |
| skeleton curve extraction | pc-skeletor Python library | initial attraction weights | 0.5 |
| | | initial contraction weights | 0.5 |
| | | point cloud down sampling distance | 0.4 m |
| skeleton point cloud downsampling | | minimum distance | 0.4 m |
| skeleton point cloud connected components | | minimum component size | 5 |
| | | octree level | 8 |
| skeleton topological reconstruction | | number of k-nearest neighbours | 12 |
| Rasterise floor point cloud | CloudCompare | pixel size | 0.04 m |
| | | pixel size | 0.04 m |

| Layer (blending mode) | parameter | value / range |
|---|---|---|
| sky-view factor (multiply) | | |
| | number of directions | 32 |
| | noise removal | 0 (none) |
| | maximum radius (pixels) | 10 |
| | linear normalisation | 0.55 – 1 |
| | opacity (%) | 25 |
| positive openness (overlay) | | |
| | number of directions | 32 |
| | noise removal | 3 (high) |
| | maximum radius (pixels) | 10 |
| | linear normalisation | $55° – 95°$ |
| | opacity (%) | 50 |
| slope gradient (luminosity) | | |
| | linear normalisation | $0° – 60°$ |
| | opacity (%) | 50 |
| hillshade (normal) | | |
| | sun elevation | 55 |
| | sun azimuth | 315 |
| | linear normalisation | 0 – 1 |
| | opacity (%) | 100 |

**Table 3.** Visual Archeology Terrain blend parameters for the cave terrain shading (Kokalj et al., 2016)

## 3.4 Floor and ceiling raster models

DEMs of the cave floor and ceiling (described in Section 2.6) are provided as a raster file in GeoTIFF format. All cave floor rasters are provided with square pixels of size 4 cm. We also include a blended image highlighting subtle topographic changes and roughness elements of the cave floor and ceiling based on the Relief Visualization Toolbox (Kokalj et al., 2016), using the presets detailed in Table 3.

## 3.5 Metadata and description

To complement the overview of notable features given below in Table 4, we also provide a set of descriptive metadata files: `cave.yaml` and `scan.yaml`. The specific details are highlighted in relevant template files (sections B1 and B2). At the

cave level, they include wherever applicable, cave entrance location and passage toponymy in relation to published maps, as well as an overview of the local geological, hydrological and speleogenetic context. For each individual scan we also give basic information about the acquisition strategy, the extent of the scanned passages with regards to the cave, and details on the instruments used and scan operators present.


**Table 4.** List of cave conduits included in the KarstConduitCatalogue

| cave | location | hydrology | bedrock lithology, *epoch* | structural setting | notable features |
|---|---|---|---|---|---|
| Archamps | Salève | small episodic stream | oolitic and/or bioclastic limestone, *Lower Cretaceous* | in overturned limb near hinge of Salève anticline, fracture controlled | paragenetic ceiling channels, solution pockets, vadose entrenchment |
| Baume de Longeaigue | Swiss Jura | epi-phreatic conduit, water exiting in springs along Le Buttes river | oolitic and pelloidal limestone, *Upper Jurassic* | in steeply dipping southern limb of Buttes anticline, bedding and fracture controlled | vertical shaft, solution pockets, floor entrenchment, small potholes |
| Grotte de la Cascade (Môtiers) | Swiss Jura | epiphreatic conduits, water flowing to Sourde spring | micritic limestone, limestone breccia and dolomite, *Upper Jurassic* | within folded beds of the Cote de Riau NW vergent anticline. Beds are vertical near the entrance and dip strongly to the SE at the upstream sump | solution pockets, ceiling channels, potholes, silt and clay deposits, eroded rimstones, collapse chambers |
| Cocalière / Cotepatière | Cévennes | mostly inactive epiphreatic gallery, water flows to Moulin de Pichegru | bioclastic and biomicritic massive white limestone, *Upper Jurassic* | within nearly flat lying beds at hinge of St André de Cruzières syncline | impacted flowstones and rimstones, potholes, cobble and gravel deposits including allochtonous elements, ceiling channel, solution pockets |
| Gouffre des Encanaux | French maritime Alps | epiphreatic gallery, linked to the Encanaux spring | biomicrite and dolomitic limestone, *Upper Jurassic* | at core of SW plunging anticline of St Baume massif, SE dipping beds visible in the cave itself | potholes, cobble and gravel deposits, ceiling channel, solution pockets |
| Event de Peyrejal | Cévennes | mostly inactive epiphreatic gallery, water flows to Moulin de Pichegru | bioclastic and biomicritic massive white limestone, *Upper Jurassic* | within nearly flat lying beds at hinge of St André de Cruzières syncline | impacted flowstones and rimstones, potholes, cobble and gravel deposits including allochtonous elements, ceiling channel, solution pockets |
| Grotte des Faux Monnayeurs | French tabular Jura | epiphreatic conduit,overflow to Pontet source | mixed bioclastic and micritic limestone, *Upper Jurassic* | along strike of steeply dipping beds, deformed by the proximity to Salinois overthrust | potholes, sparse cobble and gravel deposits, solution pockets, scallops, collapse chambers along vertical beds |
| Grotte de la Madeleine | Ardèche | inactive | bioclastic / reef limestone, Urgonian facies, *Lower Cretaceous* | horizontal, to gently SE dipping beds, undeformed | paragenetic pendants, ceiling channels, cupolas, solution pockets |

| cave | location | hydrology | bedrock lithology, *epoch* | structural setting | notable features |
|---|---|---|---|---|---|
| Grotte de la Sourde | Swiss Jura | epiphreatic conduit | micritic limestone, limestone breccia and dolomite, *Upper Jurassic* | within folded beds of the Cote de Riau NW vergent anticline. Beds are vertical to overturned | solution pockets, potholes, siphon |
| Hölloch | Swiss Alps | epiphreatic conduits | massive bioclastic limestone, *Lower Cretaceous* | NW gently dipping strata within thrust duplex of the Axen nappe | potholes, karren, solution pockets, gravel banks, erosion flutes, solution scallops |
| Lauiloch | Swiss Alps | epiphreatic conduit | massive bioclastic limestone, *Lower Cretaceous* | SE gently dipping strata within the Drusberg nappe | potholes, solution pockets, gravel banks, erosion flutes, solution scallops |
| Les Cavottes | French tabular Jura | inactive | oolithic and bioclastic limestone, *Middle Jurassic* | flat lying beds of the Ornans plateau | alteration corridors, paragenetic ceiling channels, pendants, collapse rooms |
| Markov Spodmol | Slovenian Classical Karst | stream sink | biomicrite with coal measures, *Upper Cretaceous, Paleocene* | gently NE dipping beds, NE limb of Učjenik antiform structure | solution scallops, erosion flutes and potholes, seepage karren, gravel bars, lakes |
| Rupt du Puits | Barrois | active river passage, traced to River Saulx | oolitic limestone, *Upper Jurassic* | eastern margin of Paris basin, gently SW dipping beds | lateral notches, potholes, waterfalls, solution pockets |
| Grotte de Vallorbe | Swiss Jura | epiphreatic gallery connected to the Orbe spring | bioclastic and micritic limestones, marls, *Upper Jurassic* | NW dipping beds, southern limb of a syncline, associated with NW vergent "Crêt-des-Alouettes" thrust | solution pockets and chimneys, silts to cobble bars, vadose entrenchment |
| Vers Chez Le Brandt | Swiss Jura | mostly dry small stream, dye traced to Areuse Spring | marly limestone, oolithic and oncolith well bedded limestone, *Upper Jurassic* | gently SE dipping beds on southern limb of Le Grand Bois anticline, NNW-SSE main fracture orientation | speleothems, collapse chambers |

## 4 Data examples

### 4.1 Site description

We use the example of *Markov Spodmol* (cadastral number 878), a temporary stream cave located in the classical Karst region of Slovenia to showcase the data products presented herein. The cave, with a recorded length and depth of 868 m and 61 m respectively, opens at the end of a closed valley west of Strmec mountain, at an elevation 556 m asl. The intermittent stream at the cave entrance traverses the karst massif with dye tracing connections to the Reka river.

### 4.2 Scanning procedure

The scan was carried out in May 2024, in 24 different acquisitions assembled together, totalling approximately 500 linear metres of passage, from the entrance inwards, and stopping (due to time constraints) at a 10 m pit. In parallel, a traditional speleologist's centreline was measured in order to record the geographic coordinates of 13 tie-points (Section 2.2). The entrance coordinates were derived from the Slovenian online cave cadastre.

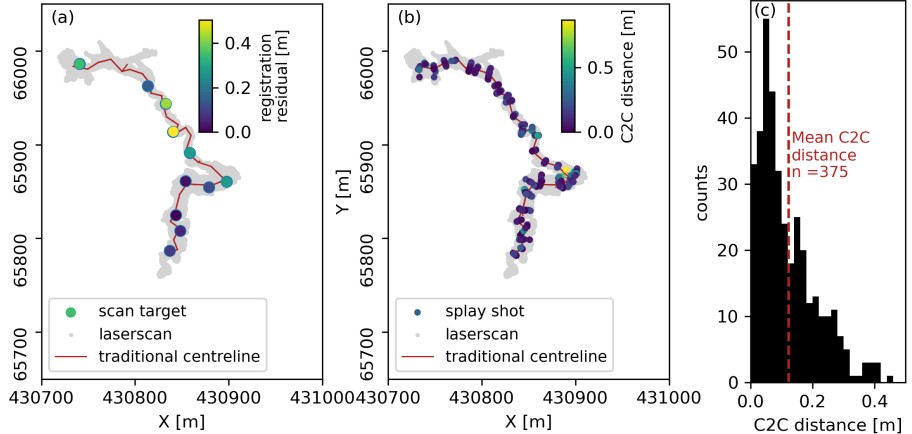

**Figure 9.** (a) Scan targets registration residuals for Markov Spodmol (b) Comparison of the traditional survey splay shots (coloured points) with the mobile laserscan (grey points) (c) Histogram of the Cloud-2-Cloud (C2C) distances calculated from splay shots to laser scan point cloud. Coordinate reference system: EPSG:3912

### 4.3 Results

#### 4.3.1 Point cloud

At Markov Spodmol, we compared two sets of point clouds collected independently: 1) a traditional set of passage dimensions from marked stations using a laser distance-metre, numbering 375 points, anchored on a centreline of 29 triplicated backwards

and forwards survey shots, and 2) a dense point cloud using the mobile BLK2GO scanner totalling a little more than $10^9$ points (Figure 8). With the mobile mapper, the effective scan time was 116 min, while the actual scan time was 240 min. As we constituted the KarstConduitCatalogue, we experienced a four-fold variation in the acquisition speed using the mobile scanner, which was highly dependent on the type of conduit. Highly convoluted passages demand that the user walk a complex trajectory in 3D to capture as many details which would otherwise be hidden, increasing the acquisition time. Nevertheless, lower and upper bounds on typical effective scan times can be given here for two typical end-members. At Hölloch, in gently inclined, tubular conduit called Riesengang whose dimensions exceed 2 m in diameter at the narrowest point, 570 m of passage were scanned in 103 min of effective scan time and 19 scenes were required altogether. The actual acquisition time including the downtime between scans, battery changes and obstacle crossing was 145 min, corresponding to a linear scanning speed of 3.9 m.min$^{-1}$. At the Baume de Longeaigue, a much more steeply inclined, convoluted cave passage with a constriction and vertical shaft, 55 m were scanned in 44 min of effective scan time and 9 scenes were required altogether. The actual acquisition time was 60 min, yielding an average scan progress speed of 0.9 m.min$^{-1}$. Therefore, at the Main Gallery of Markov Spodmol, a conduit which involved a mixture of large galleries and severe obstacles such lakes which had to be crossed by inflatable dinghy, the scanning speed of 1.7 m.min$^{-1}$ recorded falls consistently between the speeds expected for the two end-members above.

In essence, the splay shots collected during the traditional speleological survey represent a much sparser sampling of wall surface, compared to LiDAR acquisitions. The splay shots, anchored on the distoX centreline provide an independent way to check that no drift or distortion has occurred during the point cloud assembly. When measuring the centreline and repeating station sightings, mismatches arise from uncertainties in the distoX measurement due to either 1) imprecise handling of the instrument by the user 2) quality of the compass and clinometer calibration. To minimise those, we carried out triplicate station-to-station measurements, rotating the device along the sighting axis. We also measured backwards and forwards readings between any two stations. After compiling the cave survey data using the Therion software, we report an average loop error of 0.72% on the forward and backward sightings.

We provide two metrics for the comparison of distoX based surveys and the point cloud generated by laserscanning. When georeferencing the cave point cloud using the pair-wise registration method of Arun et al. (1987) on specific targets, we computed the root mean square of residuals (distances) between the two sets of points. This RMS between identified targets in the distoX survey and the laser scan was 31 cm for Markov Spodmol. Over the whole CaveConduitCatalogue, we were able to perform a similar check for drift or survey blunders on all cave datasets we intended to georeference, the RMS for target identification is given wherever applicable in Table A1. We found that this RMS of 31 cm is at the upper limit of the residual offsets observed for joint distoX and laser scanning surveys.

After georeferencing, we used CloudCompare to compute the unsigned cloud-2-cloud (C2C) distance between the traditional, sparse point cloud made of splay shots and the dense, laser-scan (Figure 9). We find that 95% of the splay end points are within a distance of 31 cm or less from the laserscan point cloud, while the mean C2C distance between the two survey techniques is 12 cm. While some splay shots may inadvertently end up close to the wall, but far from the intended corresponding features, systematic offsets due to survey blunders on the one side, or scan drift and distortion on the other, would appear in

Figure 9 as regions with either consistently high, non-random offset, or noticeable trends of increasing or decreasing offset. We could not observe any obvious spatial trend in the distribution of C2C distances on Figure 9 which would otherwise highlight first order discrepancies or major blunders between the traditional survey and the point cloud scene assembly. The agreement
between the scan and the passage dimension measurements collected using traditional speleological mapping techniques is therefore within the same error range as the registration residuals. We conclude that for the example Markov Spodmol, both survey techniques yield consistent results with respect to cave geometry at the decimetre to metre scale.

### 4.3.2 Mesh

The screened Poisson reconstruction yields a watertight surface by closing off holes in the point cloud. This results in the
erroneous reconstruction of large areas of the model using few or no data points as constraints. In addition to the cave opening, there are several large lakes in Markov Spodmol cave where no geometry data was acquired during laser scanning. We later refined the reconstructed mesh of Markov Spodmol using the mesh sculpting tool Blender to remove high uncertainty zones caused by large areas of standing water. The resulting mesh has an area of $1.5851 \times 10^4$ m$^2$ for $1.848 \times 10^7$ triangles, and the average triangle area of the reconstructed mesh is $8.6$ cm$^2$. Since the meshing procedure reconstructs the implicit surface
without honouring the data points, we calculated the cloud-to-mesh (C2M) distance using the relevant CloudCompare algorithm to detect locations where the reconstructed surface might be far from the underlying point cloud data. We find that 95% of points in the original dataset lie within $4.9$ cm of the reconstructed mesh, which is in agreement with the parameter used for the reconstruction scale in the screened Poisson Reconstruction (Table 2).

### 4.3.3 Centreline

Part of the reconstructed centreline of Markov Spodmol Cave is shown on Figure 10e-f. The centreline contains 978 points, 977 edges. There are 43 nodes of degree 1 (leaves of the tree graph, also known as external vertices) and 39 nodes of degree 3 (branch vertices). The tortuosity of individual branches, defined as $\tau_{br} = L/L_e$, with $L$ being the branch's curvilinear length and $L_e$ the euclidean distance between its start and end nodes is generally low. The mean tortuosity (weighted by branch's curvilinear length) is $\bar{\tau}_{br} = 1.17$. The arithmetic mean branch curvilinear length is $\bar{L} = 9.7$m. The downsampling of the skele-
ton point cloud with a spatial scale of 0.5 m, results in a mean edge length of 0.52 m for the centreline graph. The mode of edge length distributions is located at 0.5 m.

### 4.3.4 Raster interpretations

The extracted floor raster and combined shading allows first hand the investigation of relationships between morphological elements in the cave and an interpretation of the cave speleogenesis and sediment mobility. Figure 11 highlights the value in
culling away ceiling points to reveal the plan view morphologies of the cave passage. The floor of the entrance chamber is littered with metre scale boulders to the North, while to the South, a finer partly incised sediment bank can be seen to form a topographic step to the Southwest. Downstream of target A, a 2 m wide and several metres deep stream channel ends abruptly

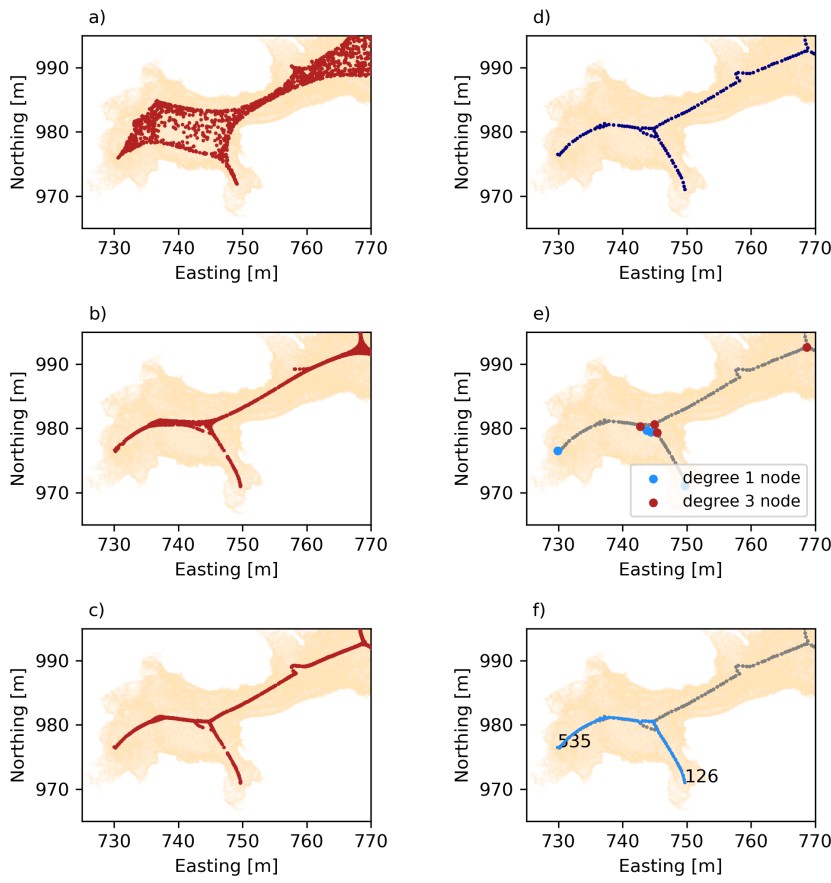

**Figure 10.** Detailed point cloud contraction and skeleton extraction workflow for the example of Markov Spodmol cave. (a-c) point cloud at different iterations of Laplacian-based contraction using the algorithm of Tagliasacchi et al. (2016). (d) spatially downsampled skeleton point cloud. (e) reconstructed Minimum Spanning Tree (MST), (f) example of a walk along the MST graph from a source to a target node. Coordinate reference system: EPSG:3912

between targets A and B, corresponding to a temporary sink. Large wooden logs are entangled around this point. A smooth, inclined bedding plane is then exposed up to target B, with a floor step corresponding to another stratum. Downstream of target

B, a karren morphology is developed for some 20 m, with two major preferential directions of development following the bedrock fractures. This gives way to a boulder-strewn passage around target C. Opposite target B, we observe a metre-high, partly incised sediment bank, deposited by slacker waters in sudden passage enlargement.

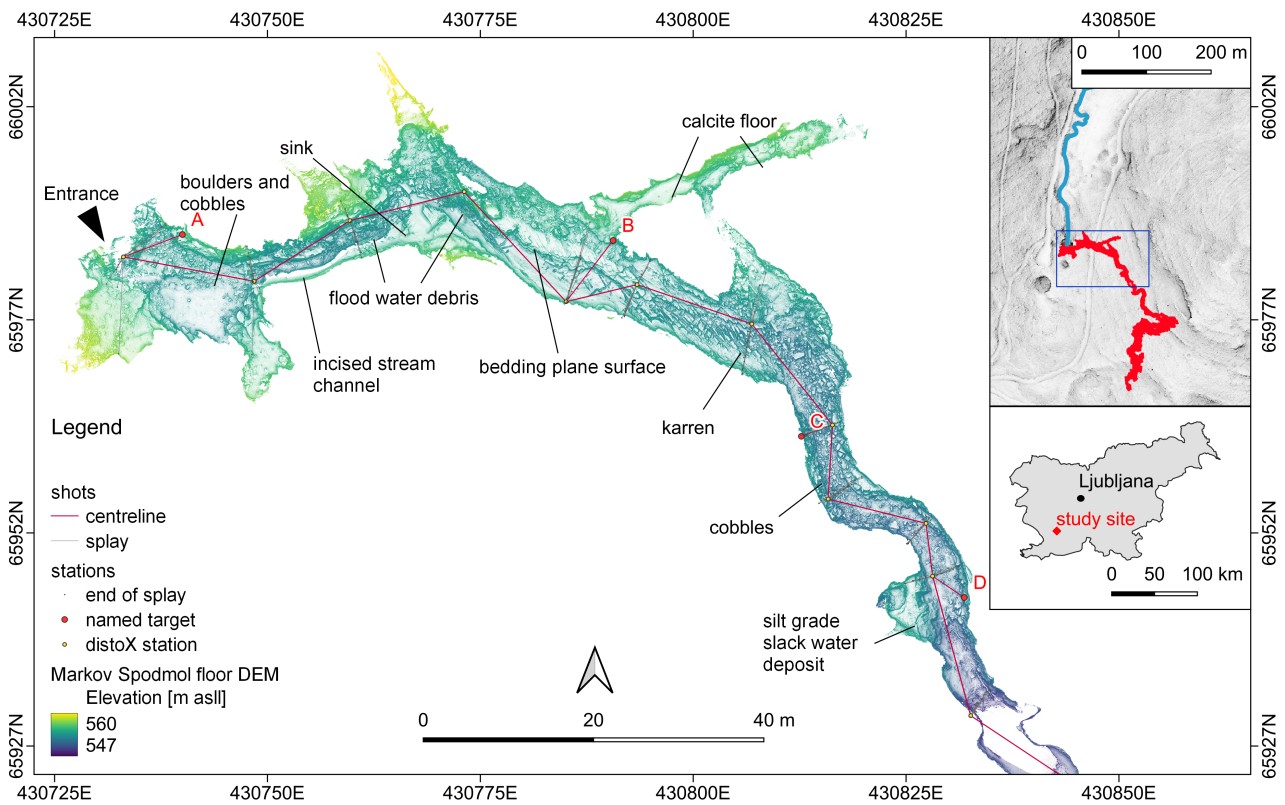

**Figure 11.** Plan view of the entrance of Markov Spodmol cave, at resolution of 4 cm per pixel, highlighting the traditional cave survey centreline used to calculate target station positions and georeference the assembled scan. *Inset*: Footprint of the cave scan with the bed of the intermittent sinking stream of Rakukic polje highlighted in blue. Coordinate reference system: EPSG:3912. Maps data: Geodetski Inštitut Slovenije © 2014 www.gis.si

## 5 Conclusions

This paper introduces a dataset of karstic conduit geometries acquired in various karst massifs around the European Alps and

beyond. They represent a spectrum of sizes, tortuosity and roughness characteristics arising from their differing host-bedrock

and speleogenetic history. The data set includes products, derived from the acquired point sets, which represent different types of generalisation of the cave conduit geometry. The triangulated meshes are reconstructions of the implicit surfaces underlying the point clouds. Raster datasets of cave floors and ceilings can be seamlessly integrated in GIS projects or databases containing other karst objects and analyse key processes controlling speleogenesis. The computed centrelines, approximations of the 3D curve skeleton of each conduit, are objects to which local geometric properties of the conduit may be attached, for instance conduit diameter, aspect ratio, shape index, etc.

The workflows presented here are specifically tailored to the cleaning and reconstruction of cave-like, georeferenced 3D objects. The point cloud cleaning to the floor / ceiling classification and rasterisation schemes depend on several parameter choices, in particular those related to spatial resampling distances, which were guided by the scanner resolution limits and the various requirements of karst geomorphology or hydraulic modelling applications. For instance, the resolution of the raster maps is in accord with the need to map decimetric objects or obstacles on the cave walls. We also demonstrate the application of automated computation of cave centrelines based on trial-and-error testing of the Laplacian-based-contraction hyper-parameters, in particular the ratio of initial contraction and attraction weights.

These numerical representations may be used to investigate a wide range of scientific questions. The raster DEM can be used for example for understanding the self-organisation of corrosion features and/or sediment deposits. The 3D point cloud can help identifying and mapping fracture orientations (e.g., Cacciari and Futai, 2017) or quantifying the geometry or density of specific geomorphological features. The unstructured point clouds acquired by laser-scanning in the underground environment usually contain gaps as well as noise for various technical or geometric reasons; they therefore present a challenge at the surface reconstruction stage. These datasets may also provide useful challenges with regards to developing semantic classification tools, as the latter could be used to segment and categorise parts of a cave point cloud as bedrock wall, secondary mineral deposit, artificial structures, etc. The surface mesh can be used for analysing, with computational fluid dynamics tools, the physical laws of water flow and solute transport in these complex geometries. They can also be used to understand typical cave geometries and relate these geometries with local geological and hydrological conditions. Finally, this work shows that the ease of use of mobile scanners allows for fast acquisition of large datasets.

*Code availability.* Data processing steps were performed using CloudCompare software and its Python wrapper CloudComPy. Centrelines were computed using the Laplacian-based contraction algorithm of Tagliasacchi et al. (2016), implemented in Python. Example scripts for the centreline extraction and rasterisation steps are given at https://github.com/ERC-Karst/pc-processing.git

*Data availability.* Data described in this manuscript can be accessed at repository under data https://doi.org/10.60544/sbjr-z851 (Racine et al., 2025a).

*Author contributions.* TR carried out fieldwork, data collection, and data curation and also wrote the manuscript. CT carried out fieldwork and data collection, curation and contributed to manuscript writing. JS developed scripts for extraction of geometrical descriptors and contributed to manuscript writing. SJ carried out field work, data acquisition and data curation. PR acquired the funding for the study to be carried out, supervised data collection and contributed to manuscript writing.

*Competing interests.* The authors declare that they have no conflict of interest.

*Disclaimer.* Any reference to specific equipment types or manufacturers is for informational purposes only and does not represent a product endorsement.

*Acknowledgements.* We are thankful for the critical help of Jürg Pulfer, Marco Filiponni, Patrick Deriaz, Alexandre Zapelli and Bertrand Hauser for organising the logistics or arranging access to many of the caves. We also acknowledge the help of Jérémie Chappuis, Robin Voland, Ana Burgoa-Tanaka, Ludovic Schorrp, Nina Egli, Ilan D'Andria, Ernesto Pugliese, Claudio Pastore, Charlotte Honiat, Amandine Laborde, Franci Gabrovšek and Matija Perne for their contribution in collecting the data in the various caves. We acknowledge funding by the European Union (ERC, KARST, 101071836). Views and opinions expressed are however those of the authors only and do not necessarily reflect those of the European Union or the European Research Council Executive Agency. Neither the European Union nor the granting authority can be held responsible for them.

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
