# Peer review of "KarstConduitCatalogue: a dataset of LiDAR derived point clouds for the analysis of karstic conduit geometry and morphology"

_Earth System Science Data, 2025_

## Referee Comment (RC1)

This work presents a valuable and timely contribution to the field of karst studies. The acquisition campaign is extensive, methodologically sound, and applied across a wide variety of cave systems in Western and Central Europe. This diversity strengthens the scientific and practical relevance of the dataset. The authors rightly highlight the potential of such data to support a range of applications, including morphological analysis, geomorphological mapping, and numerical modelling.

One of the persistent limitations of existing 3D cave surveys is that they are often conducted for a specific purpose, with limited reusability. Many datasets are eventually lost, poorly archived, or remain inaccessible to the broader community. Several studies have demonstrated that certain geometric parameters (such as network tortuosity, wall roughness…) can strongly influence physical processes, particularly flow behaviour (Albright & Springer, 2022; Kordilla et al., 2025; Peterson & Wicks, 2006). Yet these parameters are often substituted with empirical values that lack natural extrapolation. Initiatives like the one presented here offer the potential to better constrain such parameters using real-world data, and to support more robust modelling approaches in a variety of fields.

Overall, I believe this dataset represents a significant contribution and should be made available to the community. I would suggest minor revisions to improve its visibility, usability, and long-term value. The following section outlines some general criticisms and suggestions for improvement, particularly regarding the accessibility, documentation, and contextualization of the dataset.

**1. Explanation of the applications and scientific value of the dataset**

While the introduction provides a detailed overview of technical aspects related to data acquisition, it could benefit from a more explicit presentation of the scientific utility and relevance of the dataset. For instance, the authors could further develop how such 3D datasets can support geomorphological interpretations, hydrological modelling, or speleogenetic reconstructions, including references to recent studies that have leveraged or emphasized the need for similar data in karst science.

**2. Data visibility and accessibility**

Despite the quality and richness of the dataset, its visibility and long-term impact are at risk if more effective dissemination strategies are not implemented. While the complete dataset may be suitable for large-scale studies (e.g., hydrodynamic and flow regime numerical modelling), it is more realistic to expect its broader use in studies focused on a single cave. However, this presupposes that the availability and accessibility of the data are clearly highlighted and actively promoted. For that reason, two improvements seem essential:

(1) Make it possible to download data by individual cave, rather than requiring the full 200 GB dataset. As it requires more free storage than I currently have, I was unable to explore the data myself, which prevented me from analyzing it properly.

(2) Provide a geographically organized and interactive visualization interface. For example, low resolution .ply 3D previews or therion files could be archived in the Karst3D database (Karst 3D Team, 2019), which already includes an overview localisation map, access to Therion and/or low-resolution .ply 3D previews and links to full-resolution downloads and metadatas could be specified in Karst3D database.

**3. Limited contextual information on surveyed sites**

While the acquisition workflow is well described, the rationale behind the selection of specific caves remains unclear. Providing more details about the choice of surveyed sites would enhance the scientific

value and reusability of the dataset. An extended version of Table A1 could include additional descriptors for each cave, such as geographical location, geological context (e.g., lithology, structural setting), conduit morphology (e.g., phreatic, vadose..), surveyed length, and notable features. This would also help users select suitable sites for comparative or targeted analyses.

**4. SLAM acquisition – usage, limits.**

The use of mobile SLAM-based scanning is a particularly valuable aspect of this work, given the growing adoption of this technology in underground environments. However, the manuscript lacks sufficient detail on how SLAM performance and limitations were assessed. The comparison provided for a single cave is informative, but it remains unclear whether such comparisons were systematically repeated across all sites. I strongly recommend including statistics on the residual co-registration errors between DistoX measurements and the SLAM-based scans, using the target centers as a reference.

In addition, several practical aspects of the data acquisition process are missing: What were the typical acquisition times within the caves? Were the caves fully surveyed? If not, what were the limiting factors — time constraints, passage length, physical obstacles, etc.?

While the dataset is commendably diverse in terms of morphological and geological settings, it should be noted that, as I understand without having explored the data, it primarily consists of relatively horizontal cave passages (the author mention stopping the acquisition before a 10m shaft at the test cave). As such, the inherent limitations of both static and SLAM-based scanning in more complex cave sections — such as vertical shafts, wet areas, or narrow passages — should be clearly acknowledged.

Finally, to allow the reader to closely inspect the data and visually appreciate the differences between the two sources, it would be highly beneficial to include a figure showing a side-by-side comparison of a selected area (zoomed-in portion of the point clouds), along with the corresponding mesh reconstructions from TLS and PLS.

**5. Figures**

Overall, the figures are of good quality and mostly self-explanatory. However, I feel that at least one additional figure is needed to allow the reader to clearly 'see' the source data. As it stands, most illustrations are zoomed-out views of entire scanned passages, which makes it difficult to appreciate the level of detail and point cloud quality at a finer scale.

I now move on to more specific comments and minor corrections:

**Line 16:** The acronym *LIDAR* is used here for the first time but is only defined at its second occurrence at line 35.

**Line 35:** ''liDAR .. is suited to the underground as it overcomes many 35 challenges inherent to light-based techniques for the acquisition of three-dimensional point clouds (Giordan et al., 2021)''

Although the argument is valid, and could even be strengthened by mentioning the faster acquisition and post-processing times compared to visual methods, the use of this citation appears somewhat

inconsistent with the conclusions of the referenced paper. In fact, Giordan et al., (2021) highlight that visual methods, particularly Structure from Motion (SfM), offer a favourable compromise in terms of accuracy, feasibility, and cost-effectiveness for 3D surveys of complex natural caves. They emphasize that SfM constitutes a strong alternative to LiDAR, rather than being subordinate to it.

**Line 65:** The survey method (TLS, PLS) could be added in the annex table

**Line 71:** ''as well as the methods we used for scanning the cave and **processing and post-processing** the point cloud dataset''.

Seams redundant.

**Section 2.1.1** For the TLS acquisition, what was the registration method used (spheres or best fit?)

**Line 90:** ''To achieve this, the algorithm uses regular updates to the scanner position by 1) using the device's Internal Motion Unit (IMU) and 2) by triangulating between recognisable point features **(Figure 3)''**

Figure 3 shows a SLAM unit being used in a cave passage but is not really an illustration of that particular sentence regarding the SLAM method.

**Line 94:** The authors split the conduit into several overlapping acquisitions (scenes) acquired separately. It would be helpful if they could specify the criteria used for this splitting, such as whether it is based on time, length, or other factors. Furthermore, an explanation of the necessity/difficluty to perform loops and back-and-forth acquisitions would strengthen the understanding of the methodology.

**Line 98:** The sentence states that the conduit sections were scanned with a 15–35% overlap for subsequent co-registration. I understand this to mean that there is a return path of several meters or even tens of meters overlapping the previous section. However, the exact meaning and precision of these percentages in the context of field acquisition remain unclear—are these overlap values measured in real time during acquisition or calculated afterward?

**Line 114-115:** The two sentences could be combined into a single, clearer sentence to improve readability.

**Line 116, 2.2 Georeferencing:** I understand that the georeferencing was performed for all the caves, with laminated scan target as show in Fig 3 a, measured with a DistoX. It is not clear to me if the authors selected the closest point to each target center (by using the intensisty/illuminance return to clearly see the target black and white pattern?) or used another method that is less dependent to the scanning density on those targets (=acquisition distance). The authors later give some statistics about the rigid transformation for the test cave based on the splay shots but it would be helpful for the reader to give additional stats (Therion loop closure error and at least min, max and average DistoX/laser residual errors on targets), for the test cave but maybe even for the overall dataset.

Regarding the georeferencing itself, unless I missed something, it is not mentioned whether the data are shared as georeferenced point clouds and meshes, or in local coordinates with the transformation to real world parameters provided separately (e.g., in metadata). This distinction is important, as many 3D software tools do not handle large coordinate values well, and georeferenced files can be significantly larger. Clarifying this aspect in the manuscript would be useful for potential users of the dataset. My personal opinion about this is that providing data in local coordinates with georeferencing in metadata is best.

**Line 136:** The noise and related cleanings are well explained in this section but, again, a figure with a zoomed portion of the point cloud would help the reader to visualise the raw data for both techniques, as well as the noise cleaning and meshing.

**Line 159:** ''The point cloud generated by the BLK2GO device has a specific 3D structure **made of criss-crossing point trails** which originates from the scanner movement during a survey.'' Same remark, I would have appreciated a visual example of this characteristic pattern within the paper itself, rather than having to download over 200 GB of data to observe it.

**Line 204:** ''Intuitively, a non-rigid cloth is draped over the upturned point cloud, and points touching the cloth are labelled as ground category''.

This approach may work well for relatively simple topographies, but it could lead to misclassification in cases where the geometry is more complex or multivalued. Ex: In the presence of a big boulder with lower face overhanging the ''true ground'', parts of the boulder will not be labelled as ground. If the authors have considered such limitations or implemented specific strategies to address them, it would be helpful to mention it.

**Line 213:** The centreline extraction protocol is clearly presented, but the motivation for producing such data should be more explicitly stated, ideally in the introduction, and supported by relevant references (e.g., Collon et al., 2017; Jouves et al., 2017). This relates to my general comment (1), suggesting that the introduction would benefit from additional bibliography on the use and scientific value of such cave survey datasets.

**Figure 6:** Subfigures a) and b) appear somewhat redundant. Applying normal shading to subfigure a) could improve the visualization by better conveying surface orientation. In contrast, the illuminance-coloured point cloud in b) does not seem to add substantial additional information. Again, I would instead suggest replacing it with a zoomed-in view of the point cloud or mesh to provide more detailed insight into the data quality and geometry.

**Line 276:** parenthesis missing

**Line 293:** ''The scan was carried out in May 2024, in 24 different acquisitions assembled together, totalling approximately 400 linear metres of passage, from the entrance inwards, and stopping at a 10 m pit.''

Including acquisition time would help the reader assess the efficiency of the SLAM method in such setting.

**Table 3: "Visual Archeology Terrain blend** parameters for the cave terrain shading**''.** I suggest to add the citation here too: Relief Visualization toolbox (Kokalj et al., 2016)

**Line 303 305:** ''The splay shots provide an independent way to check that no drift or distortion has occurred during the point cloud assembly. After georeferencing the cave point cloud using the pair-wise registration method of Arun et al. (1987) on specific targets, we used CloudCompare to...''

One could argue that analysing the residual errors on the targets after alignment — by comparing DistoX stations to the closest corresponding target centers in the scan — would provide a more reliable basis for dataset comparison than a cloud-to-cloud comparison with the splay shot ''point cloud''. The latter is extremely low-resolution, and a splay point may lie near the laser scan purely by coincidence, without reflecting an actual spatial match. At the very least, providing statistics on the residuals at the

target locations would allow verification of whether the same ~12 cm error is observed. (see same remark above for line 116).

**Figure 7:** Same remark as above regarding the reliability of the comparison method. Additionally, the colourbar is missing for the splay shot-coloured points, which makes interpretation difficult.

**Line 308: ''on Fig.7''** could be replaced by (Fig. 7).

**Line 311 312:** ''We conclude that for the example Markov Spodmol, both survey techniques yield consistent results with respect to cave geometry at the decimetre to metre scale.''

Same remark as for line 116: if you provide statistics on the DistoX-to-laser alignment at the target centers for all caves, this would support extending the validity of the statement to the entire dataset.

Another general remark here: It is not clearly stated which method — TLS, SLAM, or DistoX — is the most accurate in terms of absolute positioning. One would intuitively expect TLS to be the most precise, followed by DistoX and then SLAM, but some clarification or reference would help support this assumption.

**Line 317:** "using the mesh sculpting tool **Blender** to remove…".

Consider rephrasing to 'using the mesh sculpting tool *in* Blender' to avoid suggesting that Blender is solely a sculpting tool.

**Figure 9:** The image appears to show a colour-coded and segmented (ground) point cloud in orthographic view, rather than a true DEM. The legend and the naming of the station symbols are somewhat unclear: What is the distinction between 'scan target' and 'marked'? Additionally, the red circles mentioned in the legend do not seem to be visible in the figure itself.

**Line 369:** ''Finally, this work shows that the ease of use of mobile scanners allows for **fast acquisition** of large datasets.''

However, the text does not provide any quantitative or comparative information to support how fast the acquisition actually is.

---

## Author Comment (AC1)

**RESPONSE TO RC1 https://doi.org/10.5194/essd-2025-194-RC1**

**1. Explanation of the applications and scientific value of the dataset**

While the introduction provides a detailed overview of technical aspects related to data acquisition, it could benefit from a more explicit presentation of the scientific utility and relevance of the dataset. For instance, the authors could further develop how such 3D datasets can support geomorphological interpretations, hydrological modelling, or speleogenetic reconstructions, including references to recent studies that have leveraged or emphasized the need for similar data in karst science.

**Response:** We agree that the introduction would benefit from expanding on the use cases of LIDAR underground where the technology was used to help with speleogenetic reconstructions and geomorphological interpretations. We propose to expand paragraph 35-45 to reflect this adding relevant references,

**1. Data visibility and accessibility**

Despite the quality and richness of the dataset, its visibility and long-term impact are at risk if more effective dissemination strategies are not implemented. While the complete dataset may be suitable for large-scale studies (e.g., hydrodynamic and flow regime numerical modelling), it is more realistic to expect its broader use in studies focused on a single cave. However, this presupposes that the availability and accessibility of the data are clearly highlighted and actively promoted. For that reason, two improvements seem essential:

(1) Make it possible to download data by individual cave, rather than requiring the full 200 GB dataset. As it requires more free storage than I currently have, I was unable to explore the data myself, which prevented me from analyzing it properly.

(2) Provide a geographically organized and interactive visualization interface. For example, low resolution .ply 3D previews or therion files could be archived in the Karst3D database (Karst 3D Team, 2019), which already includes an overview localisation map, access to Therion and/or low-resolution .ply 3D previews and links to full-resolution downloads and metadatas could be specified in Karst3D database.

**Response:** We strongly agree and propose to make the cave datasets individually available for download on the SwissUbase database. In practice, each cave will see itself attributed a single doi.

For visualisation, we also agree that to improve the visibility of the dataset, a visualisation tool is required. However, since not all datasets are georeferenced, nor are they complete scans of the caves, we propose to make available the conduit point clouds in a Potree project (see the currently available: tr1813.github.io/karstconduitcatalogue-potree/DataPaper.html). There, all 19 cave conduits are displayed as annotated low-resolution pointclouds, arrayed in a regular grid in local coordinate systems. This, we feel, allows the potential user to quickly jump from conduit to conduit to assess their differences, and provides a link to the permanent doi of each dataset.

Example of the Potree visualisation screen proposed.

Example of the Potree visualisation screen proposed, zoomed in.

**1. Limited contextual information on surveyed sites**

While the acquisition workflow is well described, the rationale behind the selection of specific caves remains unclear. Providing more details about the choice of surveyed sites would enhance the scientific value and reusability of the dataset. An extended version of Table A1 could include additional

descriptors for each cave, such as geographical location, geological context (e.g., lithology, structural setting), conduit morphology (e.g., phreatic, vadose..), surveyed length, and notable features. This would also help users select suitable sites for comparative or targeted analyses.

**Response:** We propose to add the suggested column and information in a supplementary annex table, containing information also indicated in each of the cave dataset metadata files. As suggested, this will allow potential catalogue users to quickly gauge which caves may be of interest for their use case, depending on lithology, structural settings and notable features.

**1. SLAM acquisition – usage, limits.**

The use of mobile SLAM-based scanning is a particularly valuable aspect of this work, given the growing adoption of this technology in underground environments. However, the manuscript lacks sufficient detail on how SLAM performance and limitations were assessed. The comparison provided for a single cave is informative, but it remains unclear whether such comparisons were systematically repeated across all sites. I strongly recommend including statistics on the residual co-registration errors between DistoX measurements and the SLAM-based scans, using the target centers as a reference.

In addition, several practical aspects of the data acquisition process are missing: What were the typical acquisition times within the caves? Were the caves fully surveyed? If not, what were the limiting factors — time constraints, passage length, physical obstacles, etc.?

While the dataset is commendably diverse in terms of morphological and geological settings, it should be noted that, as I understand without having explored the data, it primarily consists of relatively horizontal cave passages (the author mention stopping the acquisition before a 10m shaft at the test cave). As such, the inherent limitations of both static and SLAM-based scanning in more complex cave sections — such as vertical shafts, wet areas, or narrow passages — should be clearly acknowledged.

Finally, to allow the reader to closely inspect the data and visually appreciate the differences between the two sources, it would be highly beneficial to include a figure showing a side-by-side comparison of a selected area (zoomed-in portion of the point clouds), along with the corresponding mesh reconstructions from TLS and PLS.

**Response:** We agree and propose to include the RMS errors between DistoX scanning and SLAM cloud surveys in table A1, wherever applicable, since the DistoX survey procedure was carried out as a means of georeferencing the point clouds wherever this was necessary (caves with no known or readily available cave survey data). As requested, we will expand the results section with acquisition times for several geometric end-members to provide potential SLAM users with estimates of typical acquisition speeds based on passage geometry. For instance we will include the following the data examples point cloud subsection with

"At Markov Spodmol, we compared two sets of point clouds collected independently: 1) a traditional set of passage dimensions from marked stations using a laser distance-metre, numbering 375 points, anchored on a centreline of 29 triplicated backwards and forwards survey shots, and 2) a dense point cloud using the mobile BLK2GO scanner totalling a little more than 109 points (Figure 6). With the mobile mapper, the effective scan time was 116 mins, while the actual scan time was 240 mins. As we constituted the KarstConduitCatalogue, we experienced a four-fold variation in the acquisition speed using the mobile scanner, which was highly dependent on the type of conduit. Highly convoluted passages demand that the user walk a complex trajectory in 3D to capture as many details which would otherwise be hidden, increasing the acquisition time. Nevertheless, lower and upper bounds on typical effective scan times can be given here for two typical end members. At Hölloch, in gently inclined, tubular conduit called Riesengang whose dimensions exceed 2 m in diameter at the narrowest point, 570 m of passage were scanned in 103 mins of effective scan time; 19 scenes were required altogether. The actual acquisition time including the downtime between scans, battery changes and obstacle crossing was 145 mins, corresponding to a linear scanning speed of 3.9 m/min. At the Baume de Longeaigue, a much more steeply inclined, convoluted cave passage with a constriction and vertical shaft, 55 m were scanned in 44 mins of effective scan time; 9 scenes were required altogether. The actual acquisition time was 60 mins, yielding an average scan progress

speed of 0.9 m/min. Therefore, at the Main Gallery of Markov Spodmol, a conduit which involved a mixture of large galleries and severe obstacles (lakes which had to be crossed by inflatable dinghy), the scanning speed of 1.7 m / min recorded falls consistently between the speeds expected for the two endmembers above."

We also agree to expand the presentation of SLAM scanning strategies with regards to the practical constraints and highlight its limits. As those constraints are based on the nature of obstacles encountered in caves, we will stress the criteria for the choice of conduits size and nature of obstacles, steepness of the passage floors, etc. in the methods section 2.1.2. We will produce one additional figure zooming in on relevant details of the scanning procedure, and showcasing the texture and detail of point clouds gathered using TLD and LiDAR SLAM, as well as the texture of the mesh reconstruction.

**Figures**

Overall, the figures are of good quality and mostly self-explanatory. However, I feel that at least one additional figure is needed to allow the reader to clearly 'see' the source data. As it stands, most illustrations are zoomed-out views of entire scanned passages, which makes it difficult to appreciate the level of detail and point cloud quality at a finer scale.

**Response:** As per reply to comment above, we will produce one additional figure zooming in on relevant details of the scanning procedure, and showcasing the texture and detail of point clouds gathered using TLD and LiDAR SLAM before and after cleaning, as well as the texture of the mesh reconstruction.

Our proposed additional figure.

---

## Author Comment (AC2)

**RESPONSE TO RC2 https://doi.org/10.5194/essd-2025-194-RC2**

*Specific comments*

1. *the choice of sites studied: more details should be provided on the selection of the analysed caves. A brief description of the sites should be included, highlighting any distinctive features that justified their inclusion in the catalogue (such as morphology, type of conduit, accessibility and geological representativeness). This would clarify whether the selection was guided by specific scientific criteria or logistical considerations;*

**Response:** we agree that more contextual information on the various sites and how they fit the selection criteria for inclusion in the catalogue should be provided and propose to include an additional table summarising the notable features, geological setting and location of each site, as well as the total scanned length and the scanning instrument, BLK2GO or Faro Focus, as required.

2. *acquisition and timing: it would be appropriate to indicate the acquisition timing for each site, at least in general terms: how long it took to complete the surveys; whether the caves were surveyed in their entirety or only partially; and, if partially, the reasons for any incompleteness (e.g. accessibility limitations, environmental or technical conditions);*

**Response:** As indicated in reply to RC1 (https://doi.org/10.5194/essd-2025-194-RC1), this additional information is included for Markov Spodmol, and for two endmember caves with various geometric settings and acquisition strategies. We do this to provide upper and lower bounds of acquisition speeds for any other potential LiDAR SLAM users.

3. *Data comparison and validation: Comparing LiDAR point clouds with traditional speleological data is crucial for validation. In the case of the Markov Spodmol cave, a detailed comparison was made using splay shots. However, it is unclear whether this comparison was extended to other sites in the catalogue. If similar comparisons are available for other caves, they should be mentioned explicitly to reinforce the reliability of the entire dataset;*

**Response:** We only performed the detailed splay to point cloud comparison and validation at this specific site. For other sites, a centreline and splay shots are also available to a limited extent (e.g. Vallorbe). Wherever applicable, we will now report the RMS error on the residuals of scan target registration to provide a comparison between the distoX and LiDAR SLAM acquisitions.

4. *Redundancies: The text is generally well written, but some concepts are repeated in different sections without providing additional information. For instance, the use of the dataset for geomorphological analyses, numerical simulations, and hydrological studies is reiterated in the abstract, the introduction, section 2.1, and the conclusions.*

**Response:** the potential use cases of the dataset are indeed reiterated. In order to provide additional detail, we propose to expand the introduction to show case the value of such datasets in speleogenetic interpretation, geomorphological analyses and hydrological modelling. We will do this by referring to additional existing studies in which the use of LiDAR acquisitions is central.

- *Lines 14–16: The term 'LiDAR' is used without the acronym being defined. The definition does not appear until line 35. It is recommended that the first definition be moved to the first occurrence, as per editorial convention;*

**Response:** we will change this to define the LiDAR acronym at these lines.

- *Figure 2: The figure is useful, but rather dense, and could benefit from improved readability;*

**Response:** we agree that it is dense in its current landscape format, making the text less readable. We propose the following update to allow it to breathe.

[Figure]

- *Section 2.3: I suggest adding a comparative figure to the section, showing a point cloud before and after noise filtering. Such a visual comparison would effectively*

*highlight the impact of the cleaning processes and improve the readability of the section;*

**Response:** agreed, we append a figure highlighting the systematic and manual noise removal in a before / after image in the appendix as follows:

[Figure]

*Figure X: Illustration of the cleaning process (a) before automatic removal and manual segmentation of noise points, (b) after cleaning. In red: trails arising from the scan operator scanning themselves in the narrow passage. In blue: lone points corresponding to narrow fissures incompletely scanned.*

- *Line 151: CANUPO: The name of the algorithm is reported correctly, but is never explained. I would add a brief explanatory note: "...using the CANUPO algorithm, a supervised classifier based on multi-scale analysis of local geometry...";*

  **Response:** a good point. We will include this in the updated manuscript.

- *Figure 4: The figure is difficult to read due to the small font size used in the label texts and legend;*

  **Response:** we may increase the fontsize in the label text as follows:

[Figure]

• *Figure 6: It would be advisable to make the legend more readable by enlarging the font size. In particular, the unit of measurement should be included in the DEM legend;*

**Response:** we agree, and in keeping with our response to RC1 (https://doi.org/10.5194/essd-2025-194-RC1) we combine both panels of the figure propose to increase the fontsize as follows:

[Figure]

- *Figures 7, 8: The figures are difficult to read due to the small font size used in the label texts and legend.*

  **Response:** Figure 7 has been updated according to our response to RC1 to include a third panel and further updated here to increase the font size.

[Figure]

Figure 8 has also been updated to increase the fontsize and improve readability. Here we move legend items to the figure caption.

[Figure]

Detailed point cloud contraction and skeleton extraction workflow for the example of Markov Spodmol cave. (a-c) point cloud after 1, 2 and 4 iterations of Laplacian-based contraction using the algorithm of Tagliasacchi et al. (2016). (d) spatially downsampled skeleton point cloud. (e) reconstructed Minimum Spanning Tree (MST) with degree 1 and degree 3 nodes highlighted, (f) example of a walk between named nodes 126 and 535 along the MST graph from a source to a target node. Coordinate reference system: EPSG:3912

---

## Author Response (AR1)

**KarstConduitCatalogue: a dataset of LiDAR derived point clouds for the analysis of karstic conduit geometry and morphology**

**Tanguy Racine, Celia Trunz, Julien Straubhaar, Stéphane Jaillet, and Philippe Renard**

**RESPONSE TO RC1**

1. *Explanation of the applications and scientific value of the dataset*

*While the introduction provides a detailed overview of technical aspects related to data acquisition, it could benefit from a more explicit presentation of the scientific utility and relevance of the dataset. For instance, the authors could further develop how such 3D datasets can support geomorphological interpretations, hydrological modelling, or speleogenetic reconstructions, including references to recent studies that have leveraged or emphasized the need for similar data in karst science.*

**Response:** We agree that the introduction would benefit from expanding on the use cases of LIDAR underground where the technology was used to help with speleogenetic reconstructions and geomorphological interpretations.

**Changes: Lines 47-63**

**Data visibility and accessibility**

*Despite the quality and richness of the dataset, its visibility and long-term impact are at risk if more effective dissemination strategies are not implemented. While the complete dataset may be suitable for large-scale studies (e.g., hydrodynamic and flow regime numerical modelling), it is more realistic to expect its broader use in studies focused on a single cave. However, this presupposes that the availability and accessibility of the data are clearly highlighted and actively promoted. For that reason, two improvements seem essential:*

*(1) Make it possible to download data by individual cave, rather than requiring the full 200 GB dataset.* **As it requires more free storage than I currently have, I was unable to explore the data myself, which prevented me from analyzing it properly.**

*(2) Provide a geographically organized and interactive visualization interface. For example, low resolution .ply 3D previews or therion files could be archived in the Karst3D database (Karst 3D Team, 2019), which already includes an overview localisation map, access to Therion and/or low-resolution .ply 3D previews and links to full-resolution downloads and metadatas could be specified in Karst3D database.*

**Response:** We strongly agree and propose to make the cave datasets individually available for download on the SwissUbase database. In practice, each cave will see itself attributed a single doi.

For visualisation, we also agree that to improve the visibility of the dataset, a visualisation tool is required. However, since not all datasets are georeferenced, nor are they complete scans of the caves, we propose to make available the conduit point clouds in a Potree project (see the currently available: tr1813.github.io/karstconduitcatalogue-potree/DataPaper.html). There, all 16 caves are displayed as annotated low-resolution pointclouds, arrayed in a regular grid in local coordinate systems. This, we feel, allows the potential user to quickly jump from conduit to conduit to assess their differences, and provides a link to the permanent doi of each dataset.

**Changes: mention of this dataset at lines 76-87**

[Figure]

*Example of the Potree visualisation screen proposed.*

[Figure]

*Example of the Potree visualisation screen proposed, zoomed in.*

1. **Limited contextual information on surveyed sites**

*While the acquisition workflow is well described, the rationale behind the selection of specific caves remains unclear. Providing more details about the choice of surveyed sites would enhance the scientific value and reusability of the dataset. An extended version of Table A1 could include additional*

*descriptors for each cave, such as geographical location, geological context (e.g., lithology, structural setting), conduit morphology (e.g., phreatic, vadose..), surveyed length, and notable features. This would also help users select suitable sites for comparative or targeted analyses.*

**Response:** We propose to add the suggested column and information in a supplementary annex table, containing information also indicated in each of the cave dataset metadata files. As suggested, this will allow potential catalogue users to quickly gauge which caves may be of interest for their use case, depending on lithology, structural settings and notable features.

**Changes: table 4 at line 341 contains location, lithological, structural and hydrological information as well as notable features.**

1. ***SLAM acquisition – usage, limits.***

*The use of mobile SLAM-based scanning is a particularly valuable aspect of this work, given the growing adoption of this technology in underground environments. However, the manuscript lacks sufficient detail on how SLAM performance and limitations were assessed. The comparison provided for a single cave is informative, but it remains unclear whether such comparisons were systematically repeated across all sites. I strongly recommend including statistics on the residual co-registration errors between DistoX measurements and the SLAM-based scans, using the target centers as a reference.*

*In addition, several practical aspects of the data acquisition process are missing: What were the typical acquisition times within the caves? Were the caves fully surveyed? If not, what were the limiting factors — time constraints, passage length, physical obstacles, etc.?*

*While the dataset is commendably diverse in terms of morphological and geological settings, it should be noted that, as I understand without having explored the data, it primarily consists of relatively horizontal cave passages (the author mention stopping the acquisition before a 10m shaft at the test cave). As such, the inherent limitations of both static and SLAM-based scanning in more complex cave sections — such as vertical shafts, wet areas, or narrow passages — should be clearly acknowledged.*

*Finally, to allow the reader to closely inspect the data and visually appreciate the differences between the two sources, it would be highly beneficial to include a figure showing a side-by-side comparison of a selected area (zoomed-in portion of the point clouds), along with the corresponding mesh reconstructions from TLS and PLS.*

**Response:** We agree and propose to include the RMS errors between DistoX scanning and SLAM cloud surveys in table A1, wherever applicable, since the DistoX survey procedure was carried out as a means of georeferencing the point clouds wherever this was necessary (caves with no known or readily available cave survey data). As requested, we will expand the results section with acquisition times for several geometric end-members to provide potential SLAM users with estimates of typical acquisition speeds based on passage geometry.

**Changes: lines 358-371**

We also agree to expand the presentation of SLAM scanning strategies with regards to the practical constraints and highlight its limits. As those constraints are based on the nature of obstacles encountered in caves, we will stress the criteria for the choice of conduits size and nature of obstacles, steepness of the passage floors, etc. in the methods section 2.1.2. We will produce one additional figure zooming in on relevant details of the scanning procedure, and showcasing the texture and detail of point clouds gathered using TLD and LiDAR SLAM, as well as the texture of the mesh reconstruction.

**Changes: lines 358-371 and figure 4**

*Figures*

*Overall, the figures are of good quality and mostly self-explanatory. However, I feel that at least one additional figure is needed to allow the reader to clearly 'see' the source data. As it stands, most illustrations are zoomed-out views of entire scanned passages, which makes it difficult to appreciate the level of detail and point cloud quality at a finer scale.*

**Response:** As per reply to comment above, we will produce one additional figure zooming in on relevant details of the scanning procedure, and showcasing the texture and detail of point clouds gathered using TLD and LiDAR SLAM before and after cleaning, as well as the texture of the mesh reconstruction.

**Changes: figure 4**

*I now move on to more specific comments and minor corrections:*

*Line 16: The acronym LIDAR is used here for the first time but is only defined at its second occurrence at line 35.*

**Response:** good catch, we will change this.

**Changes: line 14**

*Line 35: ''liDAR .. is suited to the underground as it overcomes many 35 challenges inherent to light-based techniques for the acquisition of three-dimensional point clouds (Giordan et al., 2021)''*

*Although the argument is valid, and could even be strengthened by mentioning the faster acquisition and post-processing times compared to visual methods, the use of this citation appears somewhat inconsistent with the conclusions of the referenced paper. In fact, Giordan et al., (2021) highlight that visual methods, particularly Structure from Motion (SfM), offer a favourable compromise in terms of accuracy, feasibility, and cost-effectiveness for 3D surveys of complex natural caves. They emphasize that SfM constitutes a strong alternative to LiDAR, rather than being subordinate to it.*

**Response:** we propose to rephrase this sentence to make it less ambiguous relative to the citation findings and putting side by side LiDAR-based and visual methods. For instance with:

**Changes: line 34-39**

*Line 65: The survey method (TLS, PLS) could be added in the annex table*

**Response:** Agreed, we will do so.

**Changes: table A1**

*Line 71: ''as well as the methods we used for scanning the cave and **processing and post-processing** the point cloud dataset''.*

*Seams redundant.*

**Response:** yes, we will rephrase it to remove the first "processing"

**Changes: lines 100-101**

*Section 2.1.1 For the TLS acquisition, what was the registration method used (spheres or best fit?)*

**Response:** spheres were used.

**Changes: mention in figure 4a**

*Line 90: ''To achieve this, the algorithm uses regular updates to the scanner position by 1) using the device's Internal Motion Unit (IMU) and 2) by triangulating between recognisable point features* **(Figure 3)''**

*Figure 3 shows a SLAM unit being used in a cave passage but is not really an illustration of that particular sentence regarding the SLAM method.*

**Response:** the citation of figure 3 can be moved to the in-cave scanning strategy section.

**Changes: lines 133-134**

*Line 94: The authors split the conduit into several overlapping acquisitions (scenes) acquired separately. It would be helpful if they could specify the criteria used for this splitting, such as whether it is based on time, length, or other factors. Furthermore, an explanation of the necessity/difficluty to perform loops and back-and-forth acquisitions would strengthen the understanding of the methodology.*

**Response:** the criteria are two-fold and both derive from our experience using the BLK2GO scanner in the field. The real time display of the acquisition progress often becomes "laggy" or freezes entirely after more than 6-7 minutes of scanning. Since we found it essential to keep track of what had been scanned in real time, we split acquisitions in chunks no longer than this. The longer scans could also lead to failures of the SLAM algorithm, leading to hard-to-retrieve  or hard-to-clean datasets. Acquiring smaller chunks improved the resilience of our scanning strategy, since it meant we only had to rescan small conduit sections if and when needed.

**Changes: explanation of these criteria at lines 126-133**

*Line 98: The sentence states that the conduit sections were scanned with a 15–35% overlap for subsequent co-registration. I understand this to mean that there is a return path of several meters or even tens of meters overlapping the previous section. However, the exact meaning and precision of these percentages in the context of field acquisition remain unclear—are these overlap values measured in real time during acquisition or calculated afterward?*

**Response:** We agree that we should make this estimate clearer: the percentage is calculated after assembling the two scans on the register 360 software, and corresponds to the ratio of points which have a nearby neighbour in the opposite scan, to the total number of points. Since the amount of overlap is impossible to accurately determine in the field, we retraced our steps anywhere between 2-10 m to garantee that acquisitions intended to be co-registered would have enough common points.

**Changes: lines 135-137**

*Line 114-115: The two sentences could be combined into a single, clearer sentence to improve readability.*

**Response:** we will combine both sentences into one to improve readability: "We set a threshold value of d=2 mm and d=5 cm (d being the minimum distance between a point and its nearest neighbour), for high- and low-resolution point clouds, respectively."

**Changes: lines 152-154**

*Line 116, 2.2 Georeferencing: I understand that the georeferencing was performed for all the caves, with laminated scan target as show in Fig 3 a, measured with a DistoX. It is not clear to me if the authors selected the closest point to each target center (by using the intensisty/illuminance return to clearly see the target black and white pattern?) or used another method that is less dependent to the scanning density on those targets (=acquisition distance). The authors later give some statistics about the rigid transformation for the test cave based on the splay shots but it would be helpful for the reader to give additional stats (Therion loop closure error and at least min, max and average DistoX/laser residual errors on targets), for the test cave but maybe even for the overall dataset.*

*Regarding the georeferencing itself, unless I missed something, it is not mentioned whether the data are shared as georeferenced point clouds and meshes, or in local coordinates with the transformation to real world parameters provided separately (e.g., in metadata). This distinction is important, as many 3D software tools do not handle large coordinate values well, and georeferenced files can be significantly larger. Clarifying this aspect in the manuscript would be useful for potential users of the dataset. My personal opinion about this is that providing data in local coordinates with georeferencing in metadata is best.*

**Response:** As per the general comment, we propose to add georeferencing residual RMS errors associated with each scan's georeferencing, wherever applicable. Data are shared in LAS format. In this format, point coordinates are stored in a local reference frame guaranteeing small numbers, while the file header contains the transformation information necessary to convert to geographic coordinates (a "global shift"). Since this information is already encapsulated within the LAS format, we saw no utility in adding a further file with an arbitrary transformation.

**Changes: more explanation of this global shift and our choice at lines 305-306**

*Line 136: The noise and related cleanings are well explained in this section but, again, a figure with a zoomed portion of the point cloud would help the reader to visualise the raw data for both techniques, as well as the noise cleaning and meshing.*

**Response:** We propose to add this visual example, as mentioned above in an additional figure highlighting the point cloud texture close up.

**Changes: panels a and b of figure 4 added. Figure 6 shows an example of "operator noise" which was later cleaned up manually.**

*Line 159: ''The point cloud generated by the BLK2GO device has a specific 3D structure **made of criss-crossing point trails** which originates from the scanner movement during a survey.'' Same remark, I would have appreciated a visual example of this characteristic pattern within the paper itself, rather than having to download over 200 GB of data to observe it.*

**Response:** See response to comment above. We propose to add this visual example, as mentioned above in an additional figure highlighting the point cloud texture close up.

**Changes: panel b of figure 4 added.**

*Line 204: ''Intuitively, a non-rigid cloth is draped over the upturned point cloud, and points touching the cloth are labelled as ground category''.*

*This approach may work well for relatively simple topographies, but it could lead to misclassification in cases where the geometry is more complex or multivalued. Ex: In the presence of a big boulder with lower face overhanging the ''true ground'', parts of the boulder will not be labelled as ground. If the authors have considered such limitations or implemented specific strategies to address them, it would be helpful to mention it.*

**Response:** We agree that this information was missing from the methods section and propose to expand the floor- and ceiling extraction methods subsection.

**Changes: lines 246-256 contain a detailed explanation of our considerations using the CSF algorithm.**

*Line 213: The centreline extraction protocol is clearly presented, but the motivation for producing such data should be more explicitly stated, ideally in the introduction, and supported by relevant references (e.g., Collon et al., 2017; Jouves et al., 2017). This relates to my general comment (1), suggesting that the introduction would benefit from additional bibliography on the use and scientific value of such cave survey datasets.*

**Response**: we agree and will expand the introduction between lines 35-45 with these considerations to further motivate the use of centrelines and include the suggested references.

**Changes: lines 64-69**

*Figure 6: Subfigures a) and b) appear somewhat redundant. Applying normal shading to subfigure a) could improve the visualization by better conveying surface orientation. In contrast, the illuminance-coloured point cloud in b) does not seem to add substantial additional information. Again, I would instead suggest replacing it with a zoomed-in view of the point cloud or mesh to provide more detailed insight into the data quality and geometry.*

**Response:** We propose to combine both sub-panels of Figure 6 into one combined figure using elevation colouring and normals orientation shading to achieve the same effect as the bottom panel 6b.

**Changes: figure 8**

*Line 276: parenthesis missing*

**Response:** we will add the parenthesis.

*Line 293: ''The scan was carried out in May 2024, in 24 different acquisitions assembled together, totalling approximately 400 linear metres of passage, from the entrance inwards, and stopping at a 10 m pit.''*

*Including acquisition time would help the reader assess the efficiency of the SLAM method in such setting.*

**Response:** we agree and propose to expand this section and contextualising the time needed for the acquisition by using the example of acquisition times needed in two end member geometries, as mentioned in the response to general comments above.

**Changes: lines 357-371**

*Table 3: ''**Visual Archeology Terrain blend** parameters for the cave terrain shading''. I suggest to add the citation here too: Relief Visualization toolbox (Kokalj et al., 2016)*

**Response:** we will add the citation here.

**Changes: table 3 caption**

*Line 303 305: ''The splay shots provide an independent way to check that no drift or distortion has occurred during the point cloud assembly. After georeferencing the cave point cloud using the pair-wise registration method of Arun et al. (1987) on specific targets, we used CloudCompare to…''*

*One could argue that analysing the residual errors on the targets after alignment — by comparing DistoX stations to the closest corresponding target centers in the scan — would provide a more reliable basis for dataset comparison than a cloud-to-cloud comparison with the splay shot ''point cloud''. The latter is extremely low-resolution, and a splay point may lie near the laser scan purely by coincidence, without reflecting an actual spatial match. At the very least, providing statistics on the residuals at the target locations would allow verification of whether the same ~12 cm error is observed. (see same remark above for line 116).*

**Response:** we did previously consider using the scan target registration residuals and will now include them in the analysis. While here, points measured with the distoX could lie purely by coincidence next to the laser scan point cloud, statistics and visual representation of the colour coded error provides an independent way of checking whether 1) gross distoX handling errors, 2) Scanner drift, 3) registration blunders happened during the scan acquisition and assembly. Having the C2C-distances randomly distributed in space, as is shown on Figure 7a, is a robust method for excluding systematic errors, blunders or drift in the acquisition.

**Changes: lines 380-386 and lines 390-396**

**Figure 7:** *Same remark as above regarding the reliability of the comparison method. Additionally, the colourbar is missing for the splay shot-coloured points, which makes interpretation difficult.*

**Response:** We propose to adapt the figure to also show the residuals after registration in a third panel with error coded registration targets. We will update the figure to show colourbars for both panels.

**Changes: figure 9**

*Line 308: ''on Fig.7''* *could be replaced by (Fig. 7).*

*Line 311 312: ''We conclude that for the example Markov Spodmol, both survey techniques yield consistent results with respect to cave geometry at the decimetre to metre scale.''*

*Same remark as for line 116: if you provide statistics on the DistoX-to-laser alignment at the target centers for all caves, this would support extending the validity of the statement to the entire dataset.*

*Another general remark here: It is not clearly stated which method — TLS, SLAM, or DistoX — is the most accurate in terms of absolute positioning. One would intuitively expect TLS to be the most precise, followed by DistoX and then SLAM, but some clarification or reference would help support this assumption.*

**Response:** with the statistics given, as suggested in the other comments, we are able to highlight that the residual RMS for scan target registration is of 31 cm. With regards to which technique provides the best absolute positioning, the registration residuals mainly reflect the distoX measurement and handling error. Unpublished data for a short conduit section comparing TLS and SLAM acquisition by the authors, shows that the cloud to cloud differences computed between well-aligned TSL and SLAM LiDAR geometries is on the order of 1 cm. Absolute positioning is best achieved by TLS, then SLAM and finally by using the DistoX. We propose to clarify this in the text in the methods section.

*Line 317: "using the mesh sculpting tool **Blender** to remove…".*

*Consider rephrasing to 'using the mesh sculpting tool in Blender' to avoid suggesting that Blender is solely a sculpting tool.*

**Response:** this is a good distinction to make, which we will implement.

**Changes: lines 240-242**

*Figure 9: The image appears to show a colour-coded and segmented (ground) point cloud in orthographic view, rather than a true DEM. The legend and the naming of the station symbols are somewhat unclear: What is the distinction between 'scan target' and 'marked'? Additionally, the red circles mentioned in the legend do not seem to be visible in the figure itself.*

**Response:** no this is a true DEM, blended with an image generated using the Raster visualisation toolbox tool available in QGIS. We propose to update the legend to make the symbols clearer. Good catch that the scan targets appearing on the legend are absent from the map, we propose to update the map accordingly.

**Changes: figure 11**

*Line 369: ''Finally, this work shows that the ease of use of mobile scanners allows for **fast acquisition** of large datasets.''*

*However, the text does not provide any quantitative or comparative information to support how fast the acquisition actually is.*

**Response:** Agreed, see above replies to comments for the contextualisation of scanning speeds.

**Changes: as outlined in replies to comments above.**

**RESPONSE TO RC2**

*Specific comments*

1. *the choice of sites studied: more details should be provided on the selection of the analysed caves. A brief description of the sites should be included, highlighting any distinctive features that justified their inclusion in the catalogue (such as morphology, type of conduit, accessibility and geological representativeness). This would clarify whether the selection was guided by specific scientific criteria or logistical considerations;*

**Response:** we agree that more contextual information on the various sites and how they fit the selection criteria for inclusion in the catalogue should be provided and propose to include an additional table summarising the notable features, geological setting and location of each site, as well as the total scanned length and the scanning instrument, BLK2GO or Faro Focus, as required.

**Changes: table 4 contains this additional information.**

2. *acquisition and timing: it would be appropriate to indicate the acquisition timing for each site, at least in general terms: how long it took to complete the surveys; whether the caves were surveyed in their entirety or only partially; and, if partially, the reasons for any incompleteness (e.g. accessibility limitations, environmental or technical conditions);*

**Response:** As indicated in reply to RC1 (https://doi.org/10.5194/essd-2025-194-RC1), this additional information is included for Markov Spodmol, and for two endmember caves

with various geometric settings and acquisition strategies. We do this to provide upper and lower bounds of acquisition speeds for any other potential LiDAR SLAM users.

**Changes: lines 358-371**

3. *Data comparison and validation: Comparing LiDAR point clouds with traditional speleological data is crucial for validation. In the case of the Markov Spodmol cave, a detailed comparison was made using splay shots. However, it is unclear whether this comparison was extended to other sites in the catalogue. If similar comparisons are available for other caves, they should be mentioned explicitly to reinforce the reliability of the entire dataset;*

**Response:** We only performed the detailed splay to point cloud comparison and validation at this specific site. For other sites, a centreline and splay shots are also available to a limited extent (e.g. Vallorbe). Wherever applicable, we will now report the RMS error on the residuals of scan target registration to provide a comparison between the distoX and LiDAR SLAM acquisitions.

**Changes: table A1 is updated with this information wherever applicable.**

4. *Redundancies: The text is generally well written, but some concepts are repeated in different sections without providing additional information. For instance, the use of the dataset for geomorphological analyses, numerical simulations, and hydrological studies is reiterated in the abstract, the introduction, section 2.1, and the conclusions.*

**Response:** the potential use cases of the dataset are indeed reiterated. In order to provide additional detail, we propose to expand the introduction to show case the value of such datasets in speleogenetic interpretation, geomorphological analyses and hydrological modelling. We will do this by referring to additional existing studies in which the use of LiDAR acquisitions is central.

**Changes: lines 47-64**

*Technical corrections*

- *Lines 14–16: The term 'LiDAR' is used without the acronym being defined. The definition does not appear until line 35. It is recommended that the first definition be moved to the first occurrence, as per editorial convention;*

  **Response:** we will change this to define the LiDAR acronym at these lines.

**Changes: lines 14-16**

- *Figure 2: The figure is useful, but rather dense, and could benefit from improved readability;*

**Response:** we agree that it is dense in its current landscape format, making the text less readable. We propose the following update to allow it to breathe.

**Changes: figure 2**

- *Section 2.3: I suggest adding a comparative figure to the section, showing a point cloud before and after noise filtering. Such a visual comparison would effectively*

*highlight the impact of the cleaning processes and improve the readability of the section;*

**Response:** agreed, we append a figure highlighting the systematic and manual noise removal in a before / after image in the appendix as follows:

**Changes: figure 5**

- *Line 151: CANUPO: The name of the algorithm is reported correctly, but is never explained. I would add a brief explanatory note: "...using the CANUPO algorithm, a supervised classifier based on multi-scale analysis of local geometry...";*

**Response:** a good point. We will include this in the updated manuscript.

- *Figure 4: The figure is difficult to read due to the small font size used in the label texts and legend;*

**Response:** we may increase the fontsize in the label text as follows:

**Changes: figure 6**

- *Figure 6: It would be advisable to make the legend more readable by enlarging the font size. In particular, the unit of measurement should be included in the DEM legend;*

**Response:** we agree, and in keeping with our response to RC1 (https://doi.org/10.5194/essd-2025-194-RC1) we combine both panels of the figure propose to increase the font-size as follows:

**Changes: figure 8**

- *Figures 7, 8: The figures are difficult to read due to the small font size used in the label texts and legend.*

**Response:** Figure 7 has been updated according to our response to RC1 to include a third panel and further updated here to increase the font size.

Figure 8 has also been updated to increase the fontsize and improve readability. Here we move legend items to the figure caption.

**Changes: figure 9 and 10**